

# Organic Functional Groups in the Submicron Aerosol at 82.5°N from 2012 to 2014

W. Richard Leaitch[1], Lynn M. Russell[2], Jun Liu[2], Felicia Kolonjari[1], Desiree Toom[1], Lin Huang[1], Sangeeta Sharma[1], Alina Chivulescu[1], Dan Veber[1], Wendy Zhang[1]

[1] Environment and Climate Change Canada (ECCC), Toronto, ON, Canada

[2] Scripps Institution of Oceanography, University of California, San Diego, La Jolla, CA, USA

*Correspondence to*: W. Richard Leaitch (Richard.Leaitch@Canada.ca)

**Abstract.** The first multi-year contributions from organic functional groups to the Arctic submicron aerosol are documented using 126 weekly-integrated samples collected from April, 2012 to October, 2014 at the Alert

Observatory (82.45°N, 62.51°W). Results from the particle transport model FLEXPART, linear regressions among the organic and inorganic components and Positive Matrix Factorization (PMF) enable associations of organic aerosol components with source types and regions. Lower organic mass concentrations (OM) but higher ratios of OM to non-sea-salt sulphate mass concentrations (nss-$SO_4^=$) accompany smaller particles during the summer (JJA). Conversely, higher OM but lower OM/nss-$SO_4^=$ accompany larger particles during winter-spring. OM ranges from

7-463 ng m$^{-3}$, and the study average is 129 ng m$^{-3}$. The monthly maximum in OM occurs during May, one month after the peak in nss-$SO_4^=$ and two months after that of elemental carbon (EC). Winter (DJF), spring (MAM), summer and fall (SON) values of OM/nss-$SO_4^=$ are 26%, 28%, 107% and 39%, respectively, and overall about 40% of the weekly variability in the OM is associated with nss-$SO_4^=$. Respective study-averaged concentrations of alkane, alcohol, acid, amine and carbonyl groups are 57 ng m$^{-3}$, 24 ng m$^{-3}$, 23 ng m$^{-3}$, 16 ng m$^{-3}$ and 11 ng m$^{-3}$,

representing 42%, 22%, 18%, 14% and 5% of the OM, respectively. Carbonyl groups, detected mostly during spring, may have a connection with snow chemistry. The seasonally highest O/C occurs during winter (0.85) and the lowest O/C is during spring (0.51); increases in O/C are largely due to increases in alcohol groups. During winter, more than 50% of the alcohol groups are associated with primary marine emissions, consistent with Shaw et al. (2010) and Frossard et al. (2011). A secondary marine connection, rather than a primary source, is suggested for

the highest and most persistence O/C observed during the coolest and cleanest summer (2013), when alcohol and acid groups made up 63% of the OM. A secondary marine source may be a general feature of the summer OM, but higher contributions from alkane groups to OM during the warmer summers of 2012 (53%) and 2014 (50%) were likely due to increased contributions from combustion sources. Evidence for significant contributions from biomass burning (BB) was present in 4% of the weeks. During the dark months (NDJF), 29%, 28% and 14% of the nss-

$SO_4^=$, EC and OM were associated with transport times over the gas flaring region of Northern Russia and other parts of Eurasia. During spring, those percentages drop to 11% and 8% for nss-$SO_4^=$ and EC, respectively, and there is no association of OM. Large percentages of the Arctic Haze characterized at Alert likely have origins farther than 10 days transport time and may be from outside of the Eurasian region. Possible sources of unusually high nss-$SO_4^=$



and OM during September-October, 2014 are volcanic emissions or the "Smoking Hills"' area of the Northwest Territories, Canada.

## 1 Introduction

The high rate of Arctic warming (e.g. AMAP, 2015) has led to considerable interest in the role of light absorbing components of the atmospheric aerosol (e.g. Sand et al., 2015). At the same time, the net impact of the aerosol to Arctic climate has been suggested to be one of cooling (Najafi et al., 2015). With continued Arctic warming, more open water may lead to more precipitation (Kopeca et al., 2016) and industrialization of the Arctic. Correspondingly, contributions from anthropogenic and natural emissions to the Arctic aerosol are expected to change, underscoring the need to strengthen our knowledge of the Arctic aerosol in order to offer more constraints to models and enable better estimates of the impact of aerosol particles on Arctic climate now and in the future.

Anthropogenic pollution in the Arctic, or Arctic Haze, increases during winter and remains elevated until about mid-spring due to shortened transport times from southern pollution sources and reduced deposition by precipitation (Rahn et al., 1977; Shaw, 1983; Barrie, 1986; Stohl (2006); Law and Stohl, 2007; Quinn et al., 2007). Stohl (2006) showed that Arctic air near the surface spends about one week north of 80°N in the winter and two weeks north of 80°N in the summer (Stohl, 2006). Below about 2 km, Europe and Northwestern Asia, or Eurasia, is the dominant source region for pollution north of 80°N during January and February, while contributions from South/Central Asian sources become dominant above 5 km (Stohl, 2006). Simulations by Fisher et al. (2011) suggest sources of sulphate at the world's northernmost continuous aerosol observatory situated on the shore of the Arctic Ocean (Alert, Nunavut, Canada) are dominated by West Asia/Siberia, Europe, oxidation of DMS and volcanism during January and February, and by DMS oxidation, Europe, East Asia, North America and West Asia/Siberia during March and April. Simulations of black carbon (BC) by Stohl et al. (2013), Qi et al. (2017) and Xu et al. (2017) attribute 25-40% of the winter BC and about 20% of the spring BC at Alert to gas flaring in Russia with Eastern and Southern Asia making contributions to the BC of about 20% in January and about 40% in April.

The summertime Arctic aerosol has a stronger association with transport from ocean regions than continental regions (Stohl, 2006), and it was once postulated as a representation of the background aerosol of the Northern Hemisphere (Megaw and Flyger, 1973). Understanding the natural components of the Arctic aerosol is as important for climate as understanding the anthropogenic components (Carslaw et al., 2013). For example, the impact of new particle formation from natural source emissions on summertime Arctic cloud has been estimated to cool the Arctic atmosphere by 0.5 W m$^{-2}$ (Croft et al., 2016).

Since aerosol composition measurements began in the Arctic about four decades ago, when the average composition of the submicron aerosol during winter-spring was estimated to be 2 μg m$^{-3}$ of sulphate, 1 μg m$^{-3}$ of organic compounds, 0.3-0.5 μg m$^{-3}$ of BC and a few tenths of a μg m$^{-3}$ of other substances (Rahn and Heidam, 1981), sulphate and equivalent BC (BC estimated from particle light absorption) during winter-spring have declined at three of the four northernmost observatories: Alert, Nunavut; Mount Zeppelin, Svalbard; Station Nord, Greenland (Heidam et al., 1999; Hirdman et al., 2010). There have been no significant trends in either sulphate or BC at the





observatory in Barrow, Alaska (Hirdman et al., 2010). Although measurements of methane sulphonic acid (MSA) from 1980 to 2009 show no net change in MSA at Alert (Sharma et al., 2012), MSA did increase from 2000 to 2009 associated with the northward migration of the marginal ice zone (Quinn et al., 2009; Sharma et al., 2012; Laing et al., 2013). Of the four northernmost observatories, the highest MSA concentrations are measured at Mount

Zeppelin, likely due to its proximity to the waters between Greenland and Northern Europe, which are a significant source of dimethyl sulphide (DMS) from May-August (e.g. Lana et al., 2011).

Observations of organic components of the Arctic aerosol, aside from MSA, are varied in detail, location and continuity. Shaw et al. (2010) found that the total organic mass (OM) concentrations over one year at Barrow ranged from 0.07 $\mu$g m$^{-3}$ in summer to 0.43 $\mu$g m$^{-3}$ in winter. Their organic functional group (OFG) analyses

showed the winter-spring OM consisted primarily of alkane and carboxylic acid groups from combustion sources and carbohydrate-like substances hypothesized to be from sea spray in spring and frost flower formation associated with new sea ice formation in winter (Shaw et al., 2010; Russell et al., 2010). Marine sources further contribute to OM in spring and summer through emissions of biogenic volatile organic compounds or BVOCs (e.g. isoprene and terpenes) that are oxidized in the atmosphere (Fu et al., 2009a), oxygenated VOCs (OVOCs: Mungall et al., 2017)

and trimethylamines (Köllner et al., 2017) as well as from direct emissions of sea spray (Russell et al., 2010; Frossard et al., 2011; 2014). During winter and early spring, much of the OM may come from Eurasian fossil fuel sources (e.g. Behrenfeldt et al., 2008; Nguyen et al., 2013; Barrett et al., 2015), and it is mixed with sulphates and nitrates (Weinbruch et al., 2012). Organic acids and organosulphates measured in samples from Station Nord suggest that OM during October to April is from distant anthropogenic sources, whereas the year-round presence of

organic sulphates in samples collected at Mount Zeppelin, indicates contributions from local sources as well as long-range transport (Hansen et al., 2014). Tracers of secondary organic aerosol (SOA) production from BVOC oxidation have been found in the late spring and summer at Alert (Fu et al., 2009a; 2009b) and in the Arctic marine boundary layer (Hu et al., 2013). During May-September, organic acids at Station Nord suggest evidence of a relatively high biogenic influence (Hansen et al., 2014). Summer observations from ships in the central Arctic

Ocean (Chang et al., 2011) and the southeast Beaufort Sea (Kawamura et al., 2012), as well as spring samples from Alert (Fu et al., 2015) indicate OM from both marine and continental sources. Fu et al. (2013) quantified 5% of sampled OC, finding the largest organic compound class was primary saccharides from marine emissions (Russell et al., 2010) followed by secondary organic groups likely formed from the oxidation of isoprene and terpenoids. The snow pack is another potential source of organic precursors (e.g. Grannas et al., 2002; Kos et al., 2014).

Reported here are the first multi-year measurements of organic aerosol composition in combination with particle size distributions above 80$^o$N. A total of 126 weekly-averaged observations of submicron-particle chemistry and particle microphysics made at the Dr. Neil Trivett Global Atmospheric Watch observatory at Alert, Nunavut (82.5°N, 75°W; elevation 210 m-MSL; Fig. 1) from April, 2012 to October, 2014 are used to explore the relative contributions of OM to the particle size distributions, the seasonal contributions to the aerosol from OFG and offer

new observations for model evaluation. Ten-day back trajectory analyses using FLEXPART, regression analyses and Positive Matrix Factorization (PMF) enable some associations of organic aerosol components with source types and regions.



## 2 Methods

### 2.1 Instrumental Methods

Routine outdoor high-volume samples of total suspended particles have been collected at Alert for inorganic chemistry since 1980 (e.g. Barrie and Hoff, 1985); those filters are not used here. In March, 2011 two filters in

stainless steel holders were introduced inside the observatory, each set behind a cyclone with a 1 μm cut diameter: a Teflon filter for inorganic analysis, and a quartz filter for organic carbon (OC) and elemental carbon (EC). Those samples are also integrated over a week to ensure detectable levels. A number of other measurements were also introduced at Alert in March, 2011, including particle number size distributions and hourly averaged non-refractory particle composition. As a special study, weekly collections of particles smaller than 1 μm on Teflon filters were

made for OFG analysis by Fourier Transform Infrared (FTIR) spectroscopy. The OFG samples began in April, 2012 and ended in October, 2014, setting the temporal boundaries for the present work. Here only the organic aerosol from the OFG analyses is used as well as the EC from the quartz-filter analyses. OC from the quartz filters will be discussed elsewhere.

The aerosol is drawn into the laboratory through an insulated sampling stack, which is a vertically oriented

10 cm-diameter stainless steel (SS) tube. The intake is about 10 m above ground, and the flow rate is approximately 1000 L/min. The aerosol is sampled out of that tube nearer the center of the flow stream using one of six 0.95 cm SS tubes inserted about 30 cm up from the tube base. From there, ambient particles are delivered to the sampling devices mostly via stainless steel tubing with some limited use of other conductive tubing. The residence times of particles from outside to their measurement point range is less than 3 s leading to some warming of the aerosol and

reduction in relative humidity (RH).

The Teflon and quartz filters for inorganics are sampled at a flow rate of 27 L min$^{-1}$. The Teflon filters are analyzed for major inorganic ions as well as oxalate and MSA by ion chromatography (IC). Details of the analytical methods and quality control remain the same as described by Li and Barrie (1993). The quartz filters, also sampled at 27 L min$^{-1}$, are analyzed for OC and EC by thermal volatilization using three temperature steps, as discussed by

Huang et al. (2006) and Chan et al. (2010). The Teflon filters for OFG analyses were housed in a wooden box outside of the observatory at ambient temperature to reduce the potential for volatilization. For better compatibility with the other measurements, the aerosol was still sampled out of the main tube with a 0.95 cm SS tube leading from the bottom of the main tube back outside to the wooden box. Flows through the OFG were approximately 8 L min$^{-1}$, and it is assumed that the 1-2 seconds the aerosol was resident inside the tube within the laboratory was insufficient

to volatilize organic components. Also, for the relatively low ambient temperatures, the OM may be present in solid form (Zobrist et al., 2008) reducing volatilization potential. After exposure, all filters were transferred to storage vessels in a cold area of the Observatory and then stored in a freezer until shipped to ECCC in Toronto where the IC and OC/EC analyses were conducted. The OFG filters were shipped to Scripps Institution of Oceanography for the OFG analysis. All samples were shipped in insulated coolers with freezer packs to minimize potential for

volatilization and bacterial influence.



Prior to the OFG analysis by FTIR spectroscopy, the filters were equilibrated in a temperature and humidity-controlled cleanroom environment for 24 h. FTIR sample spectra were measured with a Tensor 27 spectrometer (Bruker, Billerica, MA). The spectra were baselined and fitted with peaks to identify OFG using the method described by Maria et al. (2003), Russell (2003), Russell et al. (2009) and Takahama et al. (2013). Processed

in that way, the FTIR spectra provide OFG mass concentrations, including alkane, carboxylic acid, organic hydroxyl, primary amine, carbonyl, alkene, and aromatic groups, through chemical bond-based measurements in atmospheric particles collected on a substrate (Russell et al., 2009). Alkene, aromatic, organosulphate and organonitrate groups were below detection limit for all samples. Ketone and other non-acid carbonyl group contributions are estimated from a comparison of moles of carboxylic C-OH groups and carbonyl groups quantified;

non-acid carbonyl groups (which can be present in esters, aldehydes and ketones) are determined by the moles of carbonyl present in excess of quantified moles of carboxylic C-OH groups. The moles of carboxylic C-OH and carbonyl groups for which carbonyl was not determined to be in excess had a correlation coefficient (r) of 0.84 and a regression slope of 1.0. The non-acid carbonyl is determined to be ketonic rather than aldehyde carbonyl, as absorption bands between 2700 $cm^{-1}$ and 2860 $cm^{-1}$ indicative of aldehydic hydrogen were not observed in the Alert

spectra. Further details regarding the interpretation of spectra for apportioning absorbance to moles of bond or functional group, with respective detection limits, are provided by Maria et al. (2003) and Russell et al. (2009). Estimation of mass from these quantities is based on Russell (2003), where moles of measured bonds are converted to the moles of comprising atoms, and values of OM are calculated from the sum of moles of atoms multiplied by their respective molecular weights. The uncertainty in OM has been calculated to be ±23% (Russell, 2003).

Particle size distributions from 20 nm to 500 nm diameter at Alert are measured with a TSI 3034 Scanning Mobility Particle System (SMPS). Sizing and concentrations of the SMPS are verified on site using monodisperse particles of polystyrene latex and of ammonium sulfate size selected by differential mobility using a TSI 3081 differential mobility analyzer (DMA). Particle size distributions from 500 nm to 10 □m are measured with a Grimm Model 1.109 Optical Particle Counter (OPC).

Measurements of the half-hour averaged non-refractory chemical components of particles smaller than 700 nm vacuum aerodynamic diameter (VAD) were made with an Aerodyne Research Inc. Aerosol Chemical Speciation Monitor (ACSM) (Ng et al., 2011). On-site calibrations of the ACSM are done with nearly monodisperse particles of ammonium nitrate size selected using the DMA. One hour averaged measurements of sulphate, nitrate and total organics are available only for February to November, 2013.

Vehicles normally park about 700 m from the observatory. For construction at the site or if a vehicle must drive to it, the filter sampling is turned off. Microphysical data are excluded when the wind direction is from 0° to 45° true north and for periods of short events (e.g. garbage burning at Alert station, vehicles in the vicinity) to reduce potential contamination from the Alert station approximately 8 km from the site. As shown in the supplement (Fig. S1), the impact of potential station influence on concentrations of particles from 100 nm to 500 nm is negligible and

quite small for all sizes measured with the SMPS. There is no reason to expect significant contamination of the filter samples.



### 2.2 Comparison of Instrumental Methods

Comparisons of OM and $SO_4^=$ from the filters with the ACSM are shown in Fig. S2a and S2b. OM derived from the ACSM (ACSM-OM) is limited to 23 weeks during February 15, 2013 to November 6, 2013, and there are 20 weeks with corresponding ACSM and OFG filter data. For those 20 weeks, the mean OFG-OM is 121 ng m$^{-3}$

and the mean ACSM-OM for a collection efficiency (CE) of 0.5 is 83 ng m$^{-3}$. A linear regression of ACSM-OM versus OFG-OM, forced through the origin, has a slope of 0.62 and a CoD of 0.46. Also for a CE=0.5, the mean ACSM sulphate (ACSM-SO4) is 175 ng m$^{-3}$, which is compared with filter nss-$SO_4^=$ is 444 ng m$^{-3}$. A linear regression of ACSM-SO4 vs filter sulphate through the origin has a slope of 0.43 and a CoD of 0.92.

The CE can be a large source of uncertainty. Unless strongly acidic, particles comprised of OM or $SO_4^=$ may

tend to be more solid at the lower Arctic temperatures, and that will increase the frequency of bounce off the oven before volatilization (e.g. Middlebrook et al., 2012). The ACSM will underestimate in comparison with filters cut at 1 μm diameter due to reduced particle transmission efficiency above about 500 nm geometric diameter (Liu et al., 2007), and refractory components (e.g. NaCl) go undetected. Also, whereas the filter samples are integrated over one full week with relatively little interruption, some ACSM data is interrupted over the course of a week due to

instrument problems, inlet zeroes and other disruptions of the instrument sampling line. Crystalline $(NH_4)_2SO_4$ measured with an ACSM in the laboratory has a CE closer to 0.25, which may explain the lower sulphate slope (0.43) compared with OM (0.62), assuming the two components are not uniformly internally mixed with size.

Volume estimates from the filter measurements, the SMPS (<500 nm) and the SMPS plus OPC (500-1000 nm) are compared in Fig. S2c. The volume estimates from the filter mass concentrations are calculated following Eq. (1):

Filter Volume = (OM/1.2) + (($NO_3^-$+$SO_4^=$+$NH_4^+$)/1.8) + ($Na^+$/2.16) + (EC/2.0)        (1)

where the denominators are the assumed component densities in g cm$^{-3}$. Linear regressions of the SMPS volumes and the SMPS+OPC volumes versus the filter volumes through the origin have respective slopes of 0.68 and 0.84 and CoDs of 0.87 and 0.83 (p<0.01). The differences in slopes suggest that, on average, approximately 25% of the particle volume is found in the 500-1000 nm particles. Ideally, the slope of SMPS+OPC versus filter volume should

be one. The lower value of 0.84 here may result from a number of issues, including relatively more particles larger than 1 μm sampled through the cyclones ahead of the filters versus the OPC 1 μm definition, assumption of particle sphericity for the SMPS volume estimates and the density assumptions in equation 1.

Overall, OM and $SO_4^=$ from the filters covary with the ACSM values, and the comparisons of the filter volume and microphysical volume estimates are within 20%.

### 2.3 FLEXPART and PMF

### 2.3.1 FLEXPART

The Lagrangian particle dispersion model, FLEXPART 8.2 (Stohl, 2006 and references therein), was used to construct 10-day back trajectories from the Alert station during the period of investigation (April, 2012 to October, 2014). The meteorological fields were driven with ERA-Interim reanalysis data (Dee et al., 2011) at a spatial





resolution of 1° x 1° and 60 vertical levels. Parcels were released at noon each day of the period from Alert observatory location in a 4 m layer centred at 10 m above the observatory, the approximate height of the sampling intake. The switches were set so the response function (the trajectory output) was in units of seconds and could therefore be summed over geographic regions. This method provides information on how long a parcel of air spent

over a region with no consideration for loss processes or chemical transformation. The residence times of the trajectories were aggregated by geographic region, as shown in Fig. 1, for each month of the analysis period.

**2.3.2 Positive Matrix Factorisation (PMF)**

PMF has been used with FTIR measurements of atmospheric aerosols to separate contributions from different sources at various locations from polar to equatorial regions (Russell et al., 2009; 2011). PMF of the 126 PM1 mass-

weighted FTIR spectra using scaling factor matrices calculated from baselining errors with outliers downweighted during fitting processes (as described in Russell et al., 2009) for "FPEAK" rotational values of ±1, ±0.8, ±0.6, ±0.4, ±0.2, and 0 resulted in nearly identical factors. The FPEAK value of 0 was used because it had the minimum Q/Qexpected, a mathematical diagnostic that describes the accuracy of the PMF fit (Paatero et al., 2002). Seed values of 0 to 100 (varied by 10) showed the consistency of the solutions. For 2-factor to 6-factor solutions,

Q/Qexpected decreased with increasing number of factors, indicating that the measured spectra were a better fit with more factors. However, solutions with 5 or more factors included factors that had too small a fraction of the average mass to be well represented by the 126 spectra data set (≤ 5% OM), did not correlate to any source markers, and had degenerate or unrealistic spectra. The 4-factor solution was identified as the best solution because solutions with fewer factors had higher Q/Qexpected and did not sum to reproduce the original spectra as well.

**2.4 Calculation of sea salt and non-sea salt quantities**

The factors $0.037*Na^+$ and $0.251*Na^+$ are used to remove the respective sea salt components of $SO_4^=$ and $K^+$ (e.g. Keene et al., 1986), where $Na^+$ is the total. Subsequently, a factor of $1.15*[nss-K^+]$, based on the average mass ratio of Na to K in the Earth's crust, is subtracted from the $Na^+$ to estimate a sea-salt $Na^+$ (ss-$Na^+$). No iteration is done because the average mass concentration of nss-$K^+$ was only 11% of $Na^+$.

**3 Results and Discussion**

**3.1 Filter-based chemistry and particle microphysics**

Time series of weekly-average temperature, non-sea-salt sulphate (nss-$SO_4^=$), sodium ($Na^+$), EC and OM covering April 10, 2012 to October 14, 2014 are shown in Fig. 2. Increased levels of nss-$SO_4^=$ and OM during winter-spring coincide with the lower temperatures, but only about 40% of the weekly variability in the OM is

associated with nss-$SO_4^=$ based on a power-law regression: as shown in Fig. 3 and Fig. 8a, OM is neither a linear function nss-$SO_4^=$ nor the total mass concentration.

Seasonal averages and medians of the mass concentrations of OM, functional groups, major inorganic ions and EC are given in Table 1. OM ranges from 7 ng m$^{-3}$ to 463 ng m$^{-3}$ over the entire sampling period, which corresponds



closely with observations over one year at Barrow, Alaska (Shaw et al., 2010). Average OM is 26%, 28%, 107% and 39% of the average nss-$SO_4^=$ during winter (DJF), spring (MAM), summer (JJA) and fall (SON), respectively. The springtime increase in nss-$SO_4^=$ relative to winter, as in Table 1, has been related to an increase in photochemistry during the light period (Barrie et al., 1994). EC offers no evidence for a mean increase in

combustion emissions affecting Alert from dark to light (i.e. winter to spring). Therefore, much of the nearly 70% winter-to-spring increase in OM may be due to SOA produced from VOC oxidation following polar sunrise.

From September 16, 2014 to October 13, 2014, at 226 ng m$^{-3}$ and 1005 ng m$^{-3}$ respectively, OM and nss-$SO_4^=$ were much higher than typically observed at Alert during that period (e.g. Barrie and Hoff, 1985). During the same periods in 2012 and 2013, respective OM were 77 ng m$^{-3}$ and 55 ng m$^{-3}$ and respective nss-$SO_4^=$ were 38 ng m$^{-3}$ and

55 ng m$^{-3}$. EC was not higher during September and October of 2014 (10 ng m$^{-3}$) compared with 2012 (15 ng m$^{-3}$) and 2013 (10 ng m$^{-3}$).

Seasonally, OM and nss-$SO_4^=$ were lowest during the summer, with the lowest overall during the summer of 2013 (median = 31 ng m$^{-3}$ and 33 ng m$^{-3}$, respectively). Median OM and nss-$SO_4^=$ were 64 ng m$^{-3}$ and 63 ng m$^{-3}$, respectively, during 2012 and 50 ng m$^{-3}$ and 42 ng m$^{-3}$, respectively, during 2014. The summer median and average

OM vary according to the average summer temperatures: +3.4 C for 2012; -0.4 C for 2013; +1.9 C for 2014. Such variation is consistent with a greater influence from southern latitudes that offer more temperature dependent emissions (e.g. vegetative sources or biomass burning). Median EC for the summer of 2012 (23 ng m$^{-3}$) was higher than for the summers of 2013 (4 ng m$^{-3}$) and 2014 (5 ng m$^{-3}$). Similarities in MSA and ss-$Na^+$ among the three summers (respective MSA and ss-$Na^+$ median ranges: 5-7 ng m$^{-3}$ and 2-4 ng m$^{-3}$) suggest a relative steady marine

influence with a lower sea spray component compared to the higher ss-$Na^+$ during winter.

Figure 3 shows weekly-averaged particle volume concentrations of sub-500 nm particles (SMPS only) and sub-1000 nm particles (SMPS+OPC) versus the sums of the mass concentrations of the major submicron filter constituents (OM, $NO_3^-$, nss-$SO_4^=$, $NH_4^+$, $Na^+$ and EC). Also shown are the weekly averaged volume-weighted mean diameters (VMD) for the distributions below 500 nm and the OM fraction of the total submicron filter mass

concentrations. The OM fraction is shown as an average of 10-point intervals of successive mass concentrations, due to relatively high scatter among the weekly points indicated by the error bars. The volume concentrations approach the 1.35 g cm$^{-3}$ density curve (dashed line) at lower mass concentrations as the OM fraction increases; 1.35 is the average of the densities calculated as the average of the submicron filter mass concentrations less than 0.25 mg m$^{-3}$ using equation 1. At higher mass concentrations, the sub-500 nm volume deviates increasingly from the 1.35 g cm$^{-3}$

curve due in part to the presence of increasing amounts of material in particles larger than 500 nm, confirmed by the sub-1000 nm volume points, and in part due to an increase in density mostly as a result of the higher nss-$SO_4^=$ fractions. The OM fraction appears to increase with lower volume concentrations and for volume size distributions skewed towards smaller particles: OM is a higher fraction of smaller particles in cleaner air.

### 3.2 Functional groups and oxygenation

Time series of the functional group relative contributions to OM are shown in the bottom panel of Fig. 4; OM is the sum of the five functional groups. Alkane, alcohol, acid, amine and non-acid carbonyl groups account for



42%, 22%, 18%, 14% and 5% of the overall mean OM (129 ng m$^{-3}$), respectively. The acid groups concentrations (Table 1) are consistent with Kawamura et al. (2012), who found diacid concentrations generally less than 30 ng m$^{-3}$ over the Arctic Ocean in summer. Time series of O/C and OM are shown in the top panel of Fig. 4. The O/C are calculated from the OFG as described by Russell et al. (2009). Over the limited range of ACSM data, variations in

the ratio of m/z 44 to m/z 43, which is an expression of the degree of OM oxygenation (e.g. Ng et al., 2010), are consistent (Fig. S3).

       The annual variations of OM, the functional groups, EC and nss-SO$_4^=$ are shown in Fig. 5a. Seasonal patterns of most pollutants in the Arctic are generally well known (e.g. Quinn et al., 2007). Here, the most striking observation is the maximum in OM in May, one month after that of nss-SO$_4^=$, which is one month after the

maximum in EC; the OM offset is present in each of the three springs sampled and is not a result of averaging. The peak in OM is largely a result of increases in alkane and acid groups. Fig. 5b shows monthly averaged OM plotted with the ratio of OM to nss-SO$_4^=$, ss-Na$^+$, MSA, the seasonal pattern of incident solar irradiance (as a percentage of the yearly total) and O/C (multiplied by 10). The OM peak is coincident with the maximum in MSA and decreasing ss-Na$^+$, suggesting that the May peak in OM may be partly influenced by secondary processes associated with

marine sources. The maximum in OM/nss-SO$_4^=$ occurs in August, which is also when new particle formation at Alert is a maximum (Leaitch et al., 2013; Freud et al., 2017). Seasonally averaged, O/C is highest during winter and at the upper end of values commonly observed in the atmosphere (e.g. Aiken et al., 2008). The average spring O/C of 0.51 is seasonally lowest, which would seem to contrast with the increase in photo-oxidation potential following polar sunrise. As discussed later, the peak in O/C in August is due to the summer of 2013.

**3.3 – Potential Source Regions**

       Here, air parcel times over specific regions, derived from the particle trajectory model FLEXPART, are associated with the chemical components of the particles. Times are defined as percentages per month within 100 m of the surface over a geographic region during the ten days prior to reaching Alert. The identified regions, shown in Fig. 1, were mostly selected to coincide with higher anthropogenic emissions based on the NASA OMI SO$_2$

emissions map (http://disc.sci.gsfc.nasa.gov/Aura/data-holdings/OMI; Krotkov et al., 2016) with two exceptions: Region 10 (Iceland and surrounding waters) was isolated because of emissions from volcanic fissure eruptions on Iceland, specifically the Bárðarbunga volcano during late 2014 (e.g. Gauthier et al., 2016; Schmidt et al., 2015; McCoy and Hartmann, 2015); the Canadian Northwest Territories (NWT) was included because it is a potential source of BB aerosol to Alert during summer and it includes the "Smoking Hills" region of continuously burning

lignite deposits at approximately 69.5$^o$N, 126.3$^o$W (e.g. Macdonald et al., 2016). Relative times spent over regions 1-10 are shown in Fig. 6; relative times over NWT are shown in Fig. S4.

       Regressions of chemical concentrations with time over regions were done for weekly and monthly averaged values. Fig. 7a shows time series of weekly averages of concentrations of the dominant chemical component (nss-SO$_4^=$) and times over Region 1. Region 1 is the dominant region in Fig. 6, and it includes emissions from gas flaring that are believed to be a significant source of BC to the high Arctic (e.g. Stohl et al., 2013; Sand et al., 2013; Qi et

al., 2017; Xu et al., 2017). Fig. 7b shows the monthly-averaged time series of the same two measures. Larger





variations and more offsets are apparent in the weekly averages leading to lower CoD for linear regressions: CoDs for monthly and weekly averages are given in Tables 2 and S1, respectively. The number of regressions with significance at the 95% and 90% levels is slightly higher using monthly averages (29% and 40%, respectively) compared with weekly averages (26% and 38%), and the explanation of variance is higher using monthly averages.

Lower CoDs for the weekly-averaged results are likely due to matching of one-week samples with transport that may range from a few days to more than 10 days (e.g. Qi et al., 2017). The broader patterns are better represented by the monthly averages.

Correlations of chemical components at Alert with time spent over a region are affected by many factors other than whether or not the region emits those components or its precursors. For regions with primary emissions

enhanced by wind, correlations with time may be reduced because higher winds will enhance aerosol but reduce time. For regions of significant particle precursor emissions that lead to secondary particle formation, higher wind speeds may dilute emissions potentially improving correlations. Variations in deposition will alter the association of time over a region with particle components. Also, in this case sources with injection heights above 100 m may be excluded. Still, time spent over a region offers a broad indication of its potential importance for the aerosol at Alert.

Region 1 dominates the times among the identified regions of anthropogenic emissions: Regions 1-9. Collectively, Regions 1, 4, 6 and 7 (hereafter Regions 1467), mostly covering Eurasia, comprise over 98% of those times, consistent with the analysis of Freud et al. (2017). The CoDs and significance levels for linear regressions of the major chemical components with relative time spent over Region 1 and Regions of 1467 are given in Table 2. Total months are reduced from 31 after constraining the monthly uncertainties in the FLEXPART results to less than

75%. Overall, the differences between whether the species are regressed against Region 1 or Regions 1467 are small. For all months, the highest positive correlations are for EC, nss-$K^+$ and ss-$Na^+$, and there are no correlations for OM or the functional groups except the amine groups. When the points are confined to only the periods of the dark months (NDJF) and sunlit spring months (MAM), relatively high correlations of nss-$SO_4^=$ and nss-$K^+$ emerge for both periods. Two potential sources of submicron nss-$K^+$ are mining activities in Regions 1 and 4, particularly

potash mining (e.g. Orris et al., 2014), and sweetening of gas flaring emissions by $KCO_3$. Amine solutions are also used to reduce acid gas emissions during gas flaring (e.g. Rochelle, 2009; Wu et al., 2004), and that may be factor in the modest association of amine groups with Regions 1467. The strength of the correlations of nss-$K^+$ decreases from winter to spring as the relative time over these regions decreases (Fig. 6b). Ammonium and nss-$SO_4^=$ are associated with regions 1 and 1467 during winter, but only nss-$SO_4^=$ during spring. Combined with a greater

relative increase in spring compared with nss-$SO_4^=$ (Table 1), it suggest that sources of $NH_4^+$ are much different from winter to spring. The negative correlations with MSA are not surprising considering Regions 1467 are mostly land surfaces and subsequent transport to Alert is mostly over ice; this transport pathway may be a net sink for MSA.

Significant at the 90% confidence level or higher, dark- and spring-months regressions of nss-$SO_4^=$, EC and

OM with time over Regions 1467 (Fig. S5) are used to estimate percentages of those components associated with those regions by applying the variances to the average concentrations (Table 1) after subtraction of the regression intercepts: for the dark and spring months, respectively, 271 ng m$^{-3}$ and 641 ng m$^{-3}$ for nss-$SO_4^=$, 16 ng m$^{-3}$ and 29





ng m$^{-3}$ for EC and 86 ng m$^{-3}$ for OM for the dark months. During the dark months 29% of the nss-SO$_4^=$, 28% of the EC and 14% of the OM are associated with region 1. During spring, the estimates drop to 11% and 11% for nss-SO$_4^=$ and EC, respectively, with no association for OM. These are minimum estimates since the concentrations represented by the intercepts may include contributions from Regions 1467. For comparison, simulations of BC

with FLEXPART (Stohl et al., 2013) and with GEOS-Chem (Xu et al., 2017) attribute 30-40% of the winter BC and about 20% of the spring BC at Alert to gas flaring in Russia. The current minimum estimates for EC are in reasonable agreement with the modelled estimates. In a separate study using GEOS-Chem, Qi et al. (2017) estimated that 13% of the BC at Alert during March-April, 2008 was from gas flaring in northern Russia, also in line with the present estimate.

The 11% estimate of spring nss-SO$_4^=$ associated with Regions 1467 indicates that most of nss-SO$_4^=$ likely originated from other areas, and the same can be said for OM. According to the simulations of Qi et al. (2017) and Xu et al. (2017), emissions from Eastern and Southern Asia (Regions 2, 3 and 5 here) contribute significantly to BC at Alert during winter and spring. Times spent over Regions 2, 3 and 5 are indistinguishable in Fig. 6. As suggested by Qi et al. (2017), the absence of a connection with time spent over Eastern and Southern Asia is consistent with

emissions from this region taking longer to reach Alert than the 10 days used in the present FLEXPART analysis. Longer transport times will tend to buffer variations, and OM and nss-SO$_4^=$ with origins in Eastern and Southern Asia are likely represented by the regression intercepts, forming part of the constant Arctic Haze aerosol discussed by Brock et al. (2011).

    As in Section 3.1, OM and nss-SO$_4^=$, but not EC, were unusually high from September 16, 2014 to October 13,

2014. Time spent over Region 1 stands out for September, 2014 (Fig. 6), but it is no different than September, 2012 and an increase in EC is expected if Region 1 were the main source. The low EC and high nss-SO$_4^=$/OM rules out BB. One possible source is the 'Smoking Hills' (Section 3.3) of NWT, since time spent over NWT was also anomalously high during August to October, 2014 (Fig. S4). Another possible source is the fissure eruptions of the Bárðarbunga volcano in Iceland that began in late August, 2014 and continued for some months. There is no

significant time over the Iceland region (Fig. 6), but the FLEXPART analysis may not account sufficiently for the volcano peak height of about 2 km and emissions transport could have been longer than 10 days.

### 3.4 Chemical Component Correlations

    Regressions among the OFGs, nss-SO$_4^=$, EC, ss-Na$^+$, NH$_4^+$ and nss-K$^+$ are used to further connections with possible sources. Carbonyl groups are excluded due to there being only 17 weekly-averaged samples above DL.

The regressions are done for all weeks, dark weeks, spring weeks and cleaner weeks, where the latter are defined as weeks with nss-SO$_4^=$ less than 100 ng m$^{-3}$; all cleaner weeks are during JJAS. CoDs and significance levels based on linear regressions are given in Table 3. Figure 8a shows all OM data plotted against nss-SO$_4^=$, and Fig. 8b shows OM vs nss-SO$_4^=$ for the dark and cleaner weeks; the associations are positive and significant at better than 95% confidence. However, there is no association of OM and nss-SO$_4^=$ during the spring weeks (not shown), consistent

with the discussion in Section 3.3 as well as the general pattern in Table 3. Variations in OM during spring are not




related to variations in the inorganic components and EC, but connections may be hidden within relatively large intercepts of the regressions.

The alkane and acid groups are associated with EC during the dark weeks (Table 3), but not with Regions 1 and 1467 (Table 2), implying that the variations in the alkane and acid groups and some EC derive from areas other

than Regions 1467 during the dark period. The modest correlation of alkane groups with EC during the cleaner weeks is the result of correlations during 2012 (CoD=0.18) and 2014 (CoD=0.35). There is no correlation of alkane groups and EC for the cleaner period of 2013, indicating low contributions from combustion sources. The regressions of the alkane groups with nss-$SO_4^=$ separated by years, shown in Fig. 9, offer another perspective for the cleaner weeks. The slopes and intercepts for 2012 and 2014 are relatively close, but the average concentration of

alkane groups for the summer of 2013 is approximately three times lower than the other years, despite similar ranges of nss-$SO_4^=$ (and EC). Higher ratios of alkane groups to nss-$SO_4^=$ during 2012 and 2014 suggest combustion sources, including BB, but such sources are quite episodic and the general co-variance in alkane groups and nss-$SO_4^=$ during the cleaner weeks, particularly during 2013, is not combustion related. Recent observations show SOA of marine origin and lower O/C to be a significant factor in particle growth in the summer Arctic (Willis et al.,

2017). Combined alkane groups and nss-$SO_4^=$ from marine precursor emissions might be a factor in their covariance at Alert during the cleaner months. Acid groups (not shown) exhibit similar associations with nss-$SO_4^=$ and the lowest concentrations during summer 2013.

Similar to Barrow (Shaw et al., 2010) and the open ocean east of Greenland (Frossard et al., 2011) alcohol groups are correlated with ss-$Na^+$ during the dark and cleaner weeks, but not during the spring weeks (Fig. 10). By

considering the variance in alcohol groups associated with ss-$Na^+$ concentrations above the regression intercept for the dark weeks, the fraction of alcohol groups associated with ss-$Na^+$ at Alert during the dark weeks is estimated at 54%, rising to 69% when O/C >1.

O/C is above unity for 20 of the 126 weeks (Fig. 4), 10 of which were during the dark period. Nine of the other 10 weeks were from the summer of 2013, and the average O/C for summer 2013 is 1.16 compared with 0.48

for summer 2012 and 0.45 for summer 2014. The August peak in O/C in Fig. 5b is due to the summer of 2013. Relative to OM, alcohol groups were 14%, 40% and 14% for the summers of 2012, 2013 and 2014, respectively. In Fig. 10, the alcohol groups are correlated with ss-$Na^+$ for periods with O/C >1. Alcohol groups also correlate with MSA during these times: CoD=0.85, slope=88, p<0.001 during the 10 dark weeks; CoD=0.60, slope =2.5 and p<0.02 for the nine summer weeks. For the summer of 2013, the association of alcohol groups with lower

concentrations of ss-$Na^+$ and higher concentrations of MSA suggest a stronger connection with secondary marine sources. Evidence for summertime secondary marine precursors has been found recently in the form of oxygenated VOCs from photochemical reactions at the surface of waters in the NARES Strait that divides Ellesmere Island and Greenland (Mungall et al., 2017).

Amine groups can have marine sources (Facchini et al., 2008; Köllner et al., 2017) as well as anthropogenic

sources. Here, the strongest associations of the amine groups are with ss-$Na^+$ for the dark and spring weeks (Table 3). Marine emissions, either primary or secondary, may contribute a significant fraction of the amine groups.



Modest correlations with EC and nss-$K^+$ during the cleaner weeks may indicate contributions to the amine groups from combustion sources, possibly BB.

The limited presence of carbonyl groups is consistent with observations by Kawamura and Kasukabe (1996) and Kawamura et al. (2012). Nine of the 17 weeks were during spring when gas-phase carbonyls

(formaldehyde, acetaldehyde and acetone) can exhibit a diurnal cycle related to snowpack chemistry during periods of ozone depletion (Grannas et al., 2002). During these nine weeks, the acid groups and alcohol groups were much lower fractions of the OM (2% and 11%, respectively), and six of the weeks were during periods of depleted ozone (Fig. S7): hours per week with ozone less than 50% of the mean ranged from 9% to 57%. Carbonyl compounds moving to and from the snowpack during periods of low oxidant concentrations may have contributed to carbonyl

groups in particles without increasing more oxidized groups.

Carbonyl groups can be a prominent feature of BB particles (e.g. Takahama et al., 2011). Weeks with detectable carbonyl groups and OM/nss-$SO_4^=$ exceeding 0.5 are limited to four: April 19, 2012 (OM/nss-$SO_4^=$ = 0.53), May 4, 2012 (0.57), Sept. 3, 2012 (2.9) and Aug. 1, 2014 (4.3). Considering only those four weeks, the carbonyl groups correlate strongly with EC and nss-$K^+$ (CoD>0.96 and $t_{stat}$>7), and the average functional groups

composition (45% alkane groups, 36% carbonyl groups and 9% amine groups) corresponds closely with Takahama et al. (2011). The four weeks are during periods with significant forest fire activity in the Northern Hemisphere and warmer temperatures at Alert. During April, August and September of 2012, there were a number of forest fires in Siberia (e.g. Gorchakov et al., 2014). In July, 2014 fires in Siberia (NASA Earth Observatory) and the NWT were carried north, with the NWT fires reaching at least as far as Resolute Bay, Nunavut (Köllner et al., 2017). One

other week, centred on July 20, 2012, had relatively high EC (65 ng m$^{-3}$) and OM (340 ng m$^{-3}$; 2.3 times nss-$SO_4^=$). The functional group pattern was different than the above with acid groups replacing the carbonyl groups, but this may have been a more processed BB aerosol. In summary, evidence for a significant BB influence on the Alert aerosol was found during five of the 126 weeks or 4% of the time.

### 3.5 - Positive Matrix Factorization (PMF)

PMF was applied to the OFG data to offer a further perspective on factors contributing to OM. The optimal PMF solution was four factors with an average residual of 20%. The fractional and absolute contributions by season and for the dark and cleaner periods are given in Table 4. A time series of the factor concentrations is shown in Fig. S6.

Factor 1 is labelled FFC for "Fossil Fuel Combustion", as it most strongly correlates with EC (CoD=0.29),

$NH_4^+$ (CoD=0.27) and nss-$SO_4^=$ (CoD=0.19). Factor FFC makes the overall largest contribution to OM (37%), and it is dominated by alkane and alcohol groups: 44% and 41%, respectively. Its absolute contribution to OM is largest in spring, and its relative contribution to OM is largest in summer. As above, most of the spring OM may originate either from outside of regions 1467 or beyond 10 days travel time. The high relative contribution of Factor 1 to summer OM may derive from residual spring aerosol and BB, the latter being most significant during the summer of

2012. Considering the relatively low CoD connecting this factor with EC, and the higher fraction of alkane and alcohol groups, summer marine sources may also contribute to this factor, in particular during 2013.



Factor 2 is referred to as "Sea Spray", because the highest correlations are with ss-Na$^+$ (CoD=0.28), Mg$^{++}$ (CoD=0.26) and Cl$^-$ (CoD=0.46) and the slope of the correlation with MSA is negative. It is dominated by alcohol groups (77%) with acid groups comprising 12%. Overall, Factor 2 represents 8% of the total OM. The largest seasonal contribution from Factor 2 is during winter when wind speeds are on average higher over the northern

oceans and biological productivity is lower (e.g. Lana et al., 2011). Factor 2 is consistent with earlier discussion showing alcohol groups closely correlated with ss-Na$^+$ (Fig. 10). The domination of this factor by alcohol groups is agrees with the above estimate that 54% of the alcohol groups were associated with primary marine emissions.

The third factor is labelled "Mixed". The highest correlations of Factor 3 are with two of the lowest concentration components: NO$_3^-$ (CoD=0.21) and MSA (CoD=0.19). Overall, Factor 3 represents 18% of the OM,

with the contribution reaching 25% during spring and decreasing to 13% during summer. It is comprised mostly of alkane (66%) and acid groups (24%). Alkane and acid groups are correlated with EC during the dark period (Table 3), suggesting a significant contribution from combustion sources, but variance in alkane groups during the dark months is not associated with Regions 1467 (Table 2). During the cleaner months, the alkane groups exhibit significant but weak correlations with nss-SO$_4^=$, NH$_4^+$ and nss-K$^+$, and the spring increase in alkane groups is neither

correlated with major components nor with Regions 1 and 1467. As defined, Factor 3 appears to be a mix of combustion emissions and secondary oceanic sources, transported over longer distances than those connected with Regions 1467.

The fourth factor is labelled "Secondary". It correlates predominantly with nss-SO$_4^=$ (CoD=0.40) and EC (CoD=0.17). The contribution to OM from Factor 4 is 18%, with lower variability across the seasons. At 61%, acid

groups are the largest contributor to Factor 4 followed by alkane groups at 27%. Factor 4 has the highest fractional contribution from amine groups (12%) of all factors. Many increases in Factor 4 are coincident with increases in Factor 1 (Fig. S7), but there are significant differences: Factor 4 has a lower CoD with EC, lower contributions from alcohol and alkane groups, and higher contributions from acid groups and amine groups. Amine groups show little association with Regions 1 and 1467 (Table 2), and the highest CoD for amine groups in Table 3 is with ss-Na$^+$:

recent observations suggest a secondary Arctic marine source of particulate amine coincident with but in smaller sizes than sea salt particles (Köllner et al., 2017). Also, Factor 4 is the dominant factor associated with the anomalously high nss-SO$_4^=$ during the last two months of the study, when EC was relatively low. The stronger associations of Factor 4 with nss-SO$_4^=$ and organic acid groups and weaker associations with EC and alkane groups as well as the relatively high contributions from amine groups suggest this factor is linked with secondary processes

and longer transport times.

## 4 Summary

Two and a half years (April, 2012 to October, 2014) of weekly-averaged observations of organic functional groups (OFG) were combined with observations of weekly-averaged inorganic components and aerosol particle microphysics to explore the seasonal contributions from OFGs to the submicron atmospheric aerosol and potential

sources of the OFG. These are the first multi-year observations of organic aerosol functional groups above 80$^{\circ}$N.



The study-average OM is 129 ng m$^{-3}$ with a range of 7 ng m$^{-3}$ to 463 ng m$^{-3}$, similar to OM sampled over one year at Barrow, Alaska (Shaw et al., 2010). Seasonally, OM is highest during spring at 222 ng m$^{-3}$ and lowest during summer at 65 ng m$^{-3}$. Relative to nss-SO$_4^=$, OM is 26%, 28%, 107% and 39% during winter (DJF), spring (MAM), summer (JJA) and fall (SON), respectively. Of the weekly variability in OM, 30-40% was associated with

nss-SO$_4^=$. However, during spring (MAM), there was no association between OM and nss-SO$_4^=$ suggesting that the correlations during other seasons had more to do with closeness of sources than photochemistry. That said, the maxima in both OM and nss-SO$_4^=$ occurred during spring in part due to increased photochemical potential.

Study-averaged concentrations of alkane, alcohol, acid, amine and carbonyl groups are 57 ng m$^{-3}$, 24 ng m$^{-3}$, 23 ng m$^{-3}$, 16 ng m$^{-3}$ and 11 ng m$^{-3}$, respectively. The average percentages of the weekly ratios of alkane, alcohol, acid,

amine and carbonyl groups to OM are 42%, 22%, 18%, 14% and 5%, respectively. The average O/C is 0.65 with winter O/C highest (0.85) and spring O/C lowest (0.51).

A combination of FLEXPART trajectories, linear regressions among the organic and inorganic components and PMF were used to associate the organic aerosol with potential origins with a focus on three time periods: the dark period, comprising 34 weeks during November to February, inclusive; the sunlit spring period, comprising 32 weeks

during March to May, inclusive; the cleaner period, comprising 47 weeks during June to October, inclusive, constrained to nss-SO$_4^=$ less than 100 ng m$^{-3}$. The main findings follow:

1. At Alert, OM is a higher fraction of smaller particles in cleaner air. Larger particles and a lower OM fraction are associated with higher mass concentrations. On average, particle densities are close to 1.35 g cm$^{-3}$ for smaller particles and lower mass concentrations, with higher values for larger particles and higher mass

20       concentrations.

2. The annual maximum in OM occurs in May, one month after that of nss-SO$_4^=$ and two months after the maximum in EC. The OM maximum is mostly due to increases in alkane groups and to a lesser extent acid and alcohol groups. It is coincident with the annual maximum in MSA and decreasing ss-Na$^+$. These features suggest that secondary OM from marine sources overlaps with other sources contributing OM during the spring.

3. The maximum in OM/nss-SO$_4^=$ occurred in August, coincident with new particle formation at Alert.

4. Values of O/C exceeded unity for 20 of the 126 study weeks, 10 of which were during the dark period when the fraction of alcohol groups was highest. Approximately 54% of the alcohol groups were associated with ss-Na$^+$, leading to the conclusion that higher O/C during the dark period is mainly associated with sea spray, consistent with Shaw et al. (2010) and Frossard et al. (2011).

5. Values of O/C exceeded unity for nine weeks in July-August of 2013 and were highest and most persistence during this time: average = 1.4; median=1.5. The high O/C was again due to a relative increase in alcohol groups that comprised 52% of the OM, mostly at the expense of alkane groups, while acid groups remained a similar fraction overall and relative to the other summers. As the alcohol groups were strongly associated with MSA, a secondary marine source is suggested.

6. Based on higher temperatures, higher fractions of alkane groups, higher OM and EC, as well as lower O/C (0.48-0.45), the summers of 2012 and 2014 have a greater influence from combustion sources than 2013. While





BB was a factor during 2012 and 2014, there is evidence that alkane groups and nss-SO$_4^=$ from marine precursor emissions may be generally present during the cleaner months.

7. During the dark period, 29%, 28% and 14% of nss-SO$_4^=$, EC and OM, respectively were associated with transport predominantly over the gas flaring region in Northern Russia, and Eurasia in general.

8. During the spring period, 11% and 8% of the nss-SO$_4^=$ and EC were associated with transport over the region of Northern Russia and Eurasia, with no association for OM. The difference between OM and nss-SO$_4^=$ may be due to differences from volatilization and SOA production during transport as well as potentially more OM originating from outside of Eurasia.

9. Large percentages of the Arctic Haze characterized at Alert (>60%) likely have atmospheric residence times longer than 10 days from their origin and/or are from outside of the Eurasian region.

10. In 4% of the weeks, there was evidence for a significant contribution from biomass burning (BB), coincident with forest fires in Siberia and the Canadian Northwest Territories. The frequency is low, but the impact on average summer mass concentrations can be significant.

11. Nine of the 17 weeks when carbonyl groups were measured occurred in spring when snowpack chemistry can be a significant source of gas-phase carbonyls (Grannas et al., 2002). Carbonyl compounds moving to and from the snowpack during periods of lower oxidant concentrations may increase carbonyl groups in particles without increasing more oxidized groups.

12. Unusually high nss-SO$_4^=$ and OM concentrations and relatively low EC were observed from September 16, to October 14 of 2014. Possible sources may be the 'Smoking Hills' in the Canadian NWT or volcanic emissions.

*Acknowledgements.*

We are grateful to Andrew Platt, ECCC's Arctic coordinator and manager of the Dr. Neil Trivett Global Atmospheric Watch Observatory at Alert, the Observatory Operators, and the Department of National Defence for their on-going support of the Observatory and its operations. We thank Rosa Bebi for assistance with the FTIR analyses, Elton Chan for help with FLEXPART and Betty Croft and Greg Wentworth for helpful discussion. Functional groups analysis was supported by the Grants and Contributions program of ECCC. Weekly averaged data are included at the end of the supplement.





**Figure Captions**

Fig. 1. Map showing Alert and identifying regions referenced in FLEXPART analysis: 1) North-central Russia; 2) SE Asia; 3) India; 4) Western Russia; 5) Middle East; 6) Europe; 7) NW Russia; 8) Eastern North America; 9) Canadian Oil Sands; 10) Iceland and surrounding waters; NWT) Canadian Northwest Territories.

Fig. 2. Time series of temperature and weekly-integrated mass concentrations of OM, EC, nss-$SO_4^=$ and $Na^+$ for the study period: April, 10, 2012 to October, 14, 2014.

Fig. 3. Scatter plot of volume concentrations, ratio of OM to the sum of other measured chemical components ($NO_3^-$, $SO_4^=$, $NH_4^+$, $Na^+$ and EC) and particle sizes (from SMPS) as a function of the "Sum" of all major chemical components (OM, $NO_3^-$, $SO_4^=$, $NH_4^+$, $Na^+$ and EC). Volume concentrations are shown for particles <500 nm diameter (SMPS only) and particles <1000 nm diameter (SMPS+OPC). Values of OM/Sum are averaged over successive 10 values of the 'Sum' to reduce scatter. The error bars represent the range of values of OM/Sum for each 10-point average. The dashed curve is the volume vs mass concentration for a constant density of 1.35 g cm$^{-3}$ calculated using equation 1 as the average for the lower mass concentration points (<0.25 μg m$^{-3}$).

Fig. 4. Time series of weekly-averaged OM and O/C (top panel) and percent contributions of the functional groups to OM (bottom panel).

Fig. 5. Monthly averages showing the annual variations of (a) OM, OM/nss-$SO_4^=$, ss-$Na^+$, O/C (multiplied by 10) and MSA, and (b) OM, functional group concentrations, EC and nss-$SO_4^=$. Also shown in (a) is the annual pattern of solar irradiance as a percentage of the total irradiance across a year.

Fig. 6. Monthly-averaged percentage of previous 10 days spent below 100 m over Regions 1-10 identified in Fig. 1. Results are from the FLEXPART trajectory analyses for daily air parcels arriving at Alert.

Fig. 7. (a) Time series of percentage time over Region 1 from weekly averages of FLEXPART analysis with weekly-averaged nss-$SO_4^=$ mass concentrations. (b) Time series of time over Region 1 from monthly averages of FLEXPART analysis with monthly-averaged nss-$SO_4^=$ and OM concentrations. The bars in (b) delineate the dark months (NDJF) and spring months (MAM).

Fig. 8. Regressions of OM versus nss-$SO_4^=$ for all weeks, and b) for dark weeks (during NDJF) and cleaner weeks (nss-$SO_4^=$ <100 ng m$^{-3}$). Coefficients of determination are indicated. Linear and power-law regressions are shown for all points (p<0.01), and linear regressions for the dark period (p<0.01) and the cleaner period (p<0.03).

Fig. 9. Regressions of mass concentrations of alkane functional groups with nss-$SO_4^=$ for the cleaner week during each of 2012, 2013 and 2014. (2012: p<0.1; 2013: p<0.04; 2014: p<0.07)

Fig. 10. Regressions of weekly-averaged mass concentrations of alcohol functional groups with ss-$Na^+$ for the spring weeks, dark weeks, cleaner weeks and for the 10 dark weeks with O/C>1. Results of linear regressions are indicated where significant (P<0.05). Lower plot expands the cleaner weeks and identifies the nine weeks during July and August of 2013 with O/C>1.



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



**Table 1.** Seasonal Mass Concentrations (ng m$^{-3}$); statistics based on 126 weekly concentrations*.

| Period | OM | Alkane groups | Alcohol groups | Acid groups | Amine groups | Carbonyl groups | O/C | MSA | nss-SO$_4^=$ | EC | NH$_4^+$ | ss-Na$^+$ | nss-K$^+$ |
|---|---|---|---|---|---|---|---|---|---|---|---|---|---|
| | | | | | Averages | | | | | | | | |
| Winter (DJF) | 132 | 46 | 35 | 24 | 22 | 6 | 0.85 | 0.9 | 510 | 44 | 53 | 74 | 6 |
| Spring (MAM) | 222 | 106 | 33 | 40 | 22 | 23 | 0.51 | 6.5 | 792 | 36 | 102 | 61 | 7 |
| Summer (JJA) | 65 | 31 | 12 | 14 | 6 | 2 | 0.70 | 5.4 | 60 | 12 | 17 | 3 | 1 |
| Fall (SON) | 104 | 44 | 20 | 14 | 15 | 12 | 0.59 | 2.2 | 264 | 15 | 13 | 28 | 1 |
| All | 129 | 57 | 24 | 23 | 16 | 11 | 0.65 | 4.0 | 385 | 25 | 44 | 37 | 3 |
| | | | | | Medians | | | | | | | | |
| Winter (DJF) | 120 | 23 | 27 | 10 | 20 | 0 | 0.72 | 0.8 | 536 | 22 | 47 | 65 | 6 |
| Spring (MAM) | 235 | 107 | 22 | 23 | 19 | 0 | 0.46 | 4.4 | 70 | 34 | 87 | 39 | 7 |
| Summer (JJA) | 44 | 20 | 9 | 9 | 4 | 0 | 0.49 | 5.2 | 44 | 6 | 12 | 2 | 0 |
| Fall (SON) | 74 | 34 | 10 | 9 | 8 | 0 | 0.51 | 1.7 | 88 | 13 | 7 | 5 | 0 |
| All | 88 | 37 | 13 | 11 | 9 | 0 | 0.51 | 2.5 | 192 | 16 | 27 | 15 | 2 |

*- Zero is used for samples below detection limit (BDL), including the 99 samples with carbonyl BDL. Median values of zero are below DL.

**Table 2.** Coefficients of determination (r$^2$) for particle mass concentration linear regressions with times spent over indicated group-2 regions using monthly averages. Uncertainties in time over region are <75%. Dark months are NDJF and spring months are MAM. Values in bold indicate p of slope is <0.10. Values in bold and shaded gray indicate p of slope is <0.05. A minus sign in brackets indicates a negative slope.

| Time period | | All: 25 months | All: 27 months | Dark months (8) | | Spring months (7) | |
|---|---|---|---|---|---|---|---|
| Particle species | Region | 1 | 1467 | 1 | 1467 | 1 | 1467 |
| nss-SO$_4^=$ | | **0.14** | **0.14** | **0.54** | **0.61** | **0.60** | **0.59** |
| EC | | **0.46** | **0.44** | 0.36 | **0.44** | **0.50** | **0.54** |
| ss-Na$^+$ | | **0.25** | **0.28** | 0.13 | 0.16 | 0.17 | 0.13 |
| NO$_3^-$ | | **0.13** | **0.11** | 0.02 | 0.00 | 0.04 | 0.02 |
| NH$_4^+$ | | 0.05 | 0.04 | **0.54** | **0.59** | 0.02 | 0.03 |
| nss-K$^+$ | | **0.30** | **0.25** | **0.87** | **0.77** | **0.50** | **0.54** |
| | | | | | | | |
| MSA | | **0.16(-)** | **0.22(-)** | 0.01 | 0.00 | 0.10 (-) | 0.12 (-) |
| OM | | 0.04 | 0.02 | **0.42** | **0.39** | 0.24 (-) | 0.27 (-) |
| Alkane groups | | 0.00 | 0.00 | 0.03 | 0.04 | 0.14 (-) | 0.16 (-) |
| Alcohol groups | | 0.11 | **0.11** | **0.44** | 0.32 | 0.26 (-) | 0.28 (-) |
| Acid groups | | 0.02 | 0.00 | 0.19 | 0.20 | 0.42 (-) | 0.45 (-) |
| Amine groups | | **0.15** | **0.15** | 0.25 | 0.37 | 0.00 | 0.00 |





**Table 3.** Coefficients of Determination for Linear Regressions based on weekly-averaged samples. Values in bold indicate p of slope is <0.10. Shaded gray boxes indicate p of slope is <0.05. A minus sign in brackets indicates a negative slope.

| Species | OM | Alkane groups | Alcohol groups | Acid groups | Amine groups | O/C | EC | ss-Na$^+$ | NH$_4^+$ | nss-K$^+$ | MSA |
|---|---|---|---|---|---|---|---|---|---|---|---|
| | | | | | All weeks (125, except for values with EC based on 119) | | | | | | |
| nss-SO$_4^=$ | **0.30** | **0.22** | **0.04** | **0.03** | **0.32** | **0.03(-)** | 0.30 | **0.20** | **0.48** | **0.64** | 0.01 |
| EC | **0.23** | **0.22** | 0.00 | **0.18** | **0.11** | **0.02(-)** | | **0.01** | **0.30** | **0.54** | 0.00 |
| ss-Na$^+$ | **0.08** | 0.01 | **0.22** | 0.00 | **0.30** | **0.03** | | | **0.10** | **0.17** | 0.00 |
| NH$_4^+$ | **0.25** | **0.26** | **0.02** | **0.10** | **0.09** | **0.04** | | | | **0.56** | **0.03** |
| nss-K$^+$ | **0.23** | **0.17** | **0.05** | **0.08** | **0.14** | 0.01(-) | | | | | 0.00 |
| MSA | **0.08** | **0.14** | 0.00 | **0.09** | 0.01 | **0.05(-)** | | | | | |
| | | | | | 34 Dark weeks (NDJF) | | | | | | |
| nss-SO$_4^=$ | **0.31** | **0.27** | 0.01 | **0.12** | **0.22** | **0.08(-)** | 0.56 | 0.02 | **0.72** | **0.63** | **0.36** |
| EC | **0.41** | **0.62** | 0.01(-) | **0.60** | **0.11** | **0.16(-)** | | 0.03 | **0.78** | **0.56** | **0.28** |
| ss-Na$^+$ | **0.09** | 0.02(-) | **0.57** | 0.03(-) | **0.39** | **0.18** | | | 0.00 | 0.00 | **0.12** |
| NH$_4^+$ | **0.42** | **0.41** | 0.01 | **0.43** | **0.20** | **0.08(-)** | | | | **0.80** | **0.38** |
| nss-K$^+$ | **0.38** | **0.28** | 0.04 | **0.21** | **0.21** | 0.01(-) | | | | | **0.23** |
| MSA | **0.40** | **0.32** | **0.11** | **0.16** | **0.20** | **0.11(-)** | | | | | |
| | | | | | 32 Spring weeks (MAM) | | | | | | |
| nss-SO$_4^=$ | 0.03(-) | 0.06(-) | 0.01(-) | **0.13(-)** | 0.01 | 0.00 | **0.24** | **0.12** | **0.16** | **0.62** | 0.03(-) |
| EC | 0.02 | 0.01 | 0.01(-) | 0.01(-) | 0.01 | 0.00 | | 0.08(-) | 0.05 | **0.53** | 0.07(-) |
| ss-Na$^+$ | 0.03(-) | 0.04(-) | 0.01(-) | 0.02(-) | **0.17** | 0.01(-) | | | 0.00 | 0.01 | 0.02 |
| NH$_4^+$ | 0.00 | 0.00 | 0.02(-) | 0.03(-) | 0.01(-) | 0.02(-) | | | | **0.11** | 0.01(-) |
| nss-K$^+$ | 0.04(-) | 0.06(-) | 0.03(-) | 0.05(-) | 0.01(-) | 0.00 | | | | | **0.12(-)** |
| MSA | 0.07 | **0.12** | 0.01(-) | **0.10** | 0.09 | 0.03(-) | | | | | |
| | | | | | 47 Cleaner weeks (JJASO and nss-SO$_4^=$ <100 ng m$^{-3}$)* | | | | | | |
| nss-SO$_4^=$ | **0.10** | **0.16** | **0.06** | **0.10** | 0.00 | **0.11(-)** | 0.11 | 0.02 | **0.43** | **0.25** | **0.09** |
| EC | **0.19** | **0.18** | 0.02 | **0.07** | **0.07** | 0.06(-) | | 0.00 | **0.19** | **0.28** | 0.00 |
| ss-Na$^+$ | 0.02 | 0.00 | **0.45** | 0.00 | 0.02 | 0.01 | | | 0.02(-) | 0.00 | 0.00 |
| NH$_4^+$ | **0.08** | **0.12** | 0.01 | 0.05 | 0.00 | **0.08(-)** | | | | **0.52** | **0.33** |
| nss-K$^+$ | **0.25** | **0.21** | 0.03 | 0.04 | **0.15** | 0.05(-) | | | | | **0.20** |
| MSA | 0.02 | 0.02 | 0.01 | 0.01 | 0.00 | 0.00 | | | | | |

**Table 4.** Mean percentage contributions to OM from PMF factors (126 samples). The difference from 100% of the sum of factors is residual (ca. 20%). Values in parentheses are the factor mass concentrations in ng m$^{-3}$.

| Seasonal Period | % Factor 1 "FFC" (ng m$^{-3}$) | % Factor 2 "Sea spray" (ng m$^{-3}$) | % Factor 3 "Mixed" (ng m$^{-3}$) | % Factor 4 "Secondary" (ng m$^{-3}$) |
|---|---|---|---|---|
| Winter (DJF) | 23 (39) | 17 (22) | 18 (29) | 18 (22) |
| Spring (MAM) | 37 (68) | 3 (8) | 25 (63) | 18 (31) |
| Summer (JJA) | 47 (24) | 6 (3) | 13 (14) | 12 (9) |
| Fall (SON) | 35 (28) | 7 (11) | 18 (19) | 18 (22) |
| All | 37 (40) | 8 (10) | 18 (31) | 17 (20) |
| Dark months (NDJF) | 24 (38) | 17 (25) | 18 (31) | 18 (21) |
| Cleaner months (JJASO) | 42 (21) | 5 (3) | 17 (9) | 16 (10) |





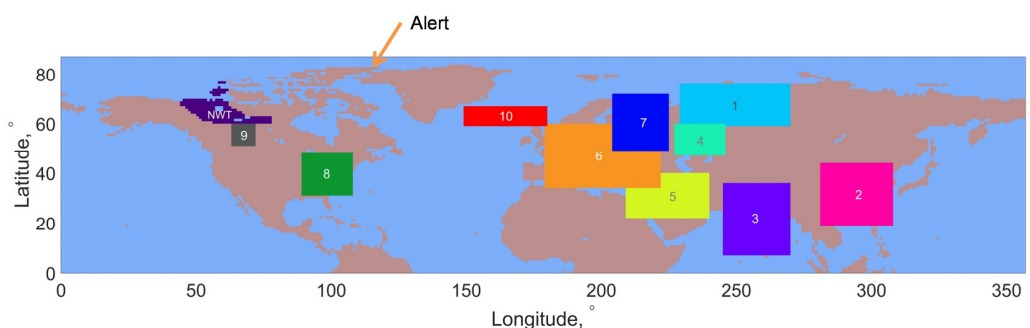

**Figure 1. Map showing Alert and identifying regions referenced in FLEXPART analysis: 1) North-central Russia; 2) SE Asia; 3) India; 4) Western Russia; 5) Middle East; 6) Europe; 7) NW Russia; 8) Eastern North America; 9) Canadian Oil Sands; 10) Iceland and surrounding waters; NWT) Canadian Northwest Territories.**

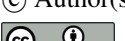



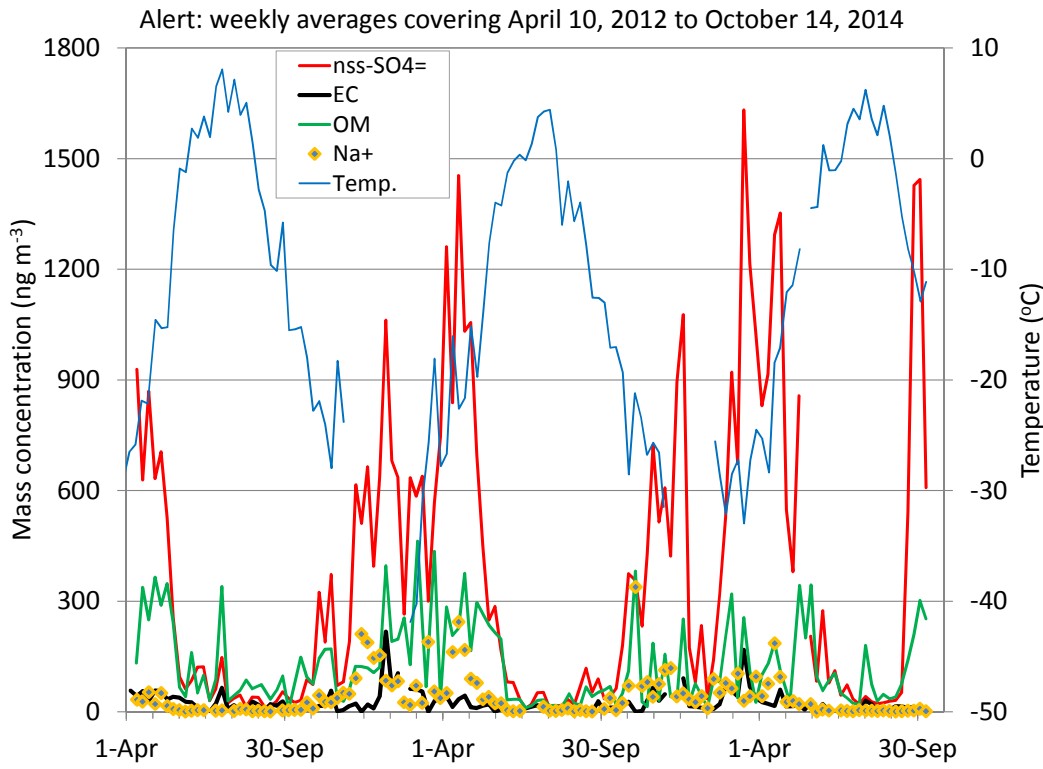

**Figure 2. Time series of temperature and weekly-integrated mass concentrations of OM, EC, nss-SO$_4^=$ and Na$^+$ for the**
5   **study period: April, 10, 2012 to October, 14, 2014.**





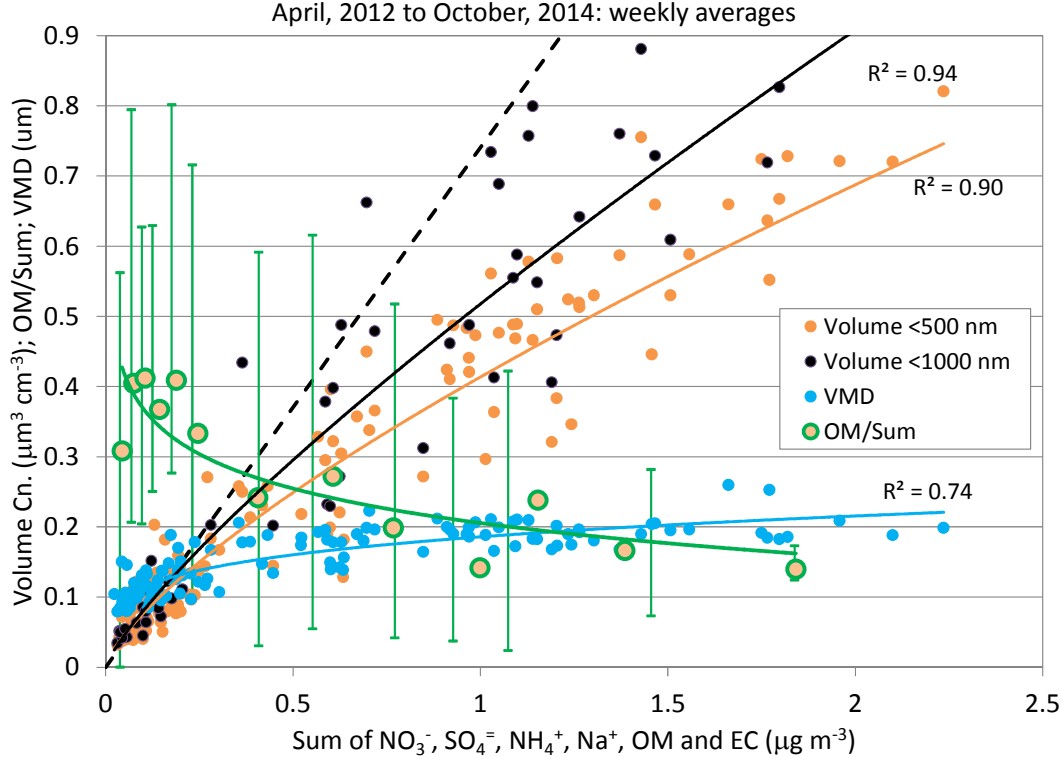

**Figure 3. Scatter plot of volume concentrations, ratio of OM to the sum of other measured chemical components (NO$_3^-$, SO$_4^=$, NH$_4^+$, Na$^+$ and EC) and particle sizes (from SMPS) as a function of the "Sum" of all major chemical components (OM, NO$_3^-$, SO$_4^=$, NH$_4^+$, Na$^+$ and EC). Volume concentrations are shown for particles <500 nm diameter (SMPS only) and particles <1000 nm diameter (SMPS+OPC). Values of OM/Sum are averaged over successive 10 values of the 'Sum' to reduce scatter. The error bars represent the range of values of OM/Sum for each 10-point average. The dashed curve is the volume vs mass concentration for a constant density of 1.35 g cm$^{-3}$ calculated using equation 1 as the average for the lower mass concentration points (<0.25 µg m$^{-3}$).**

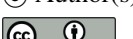



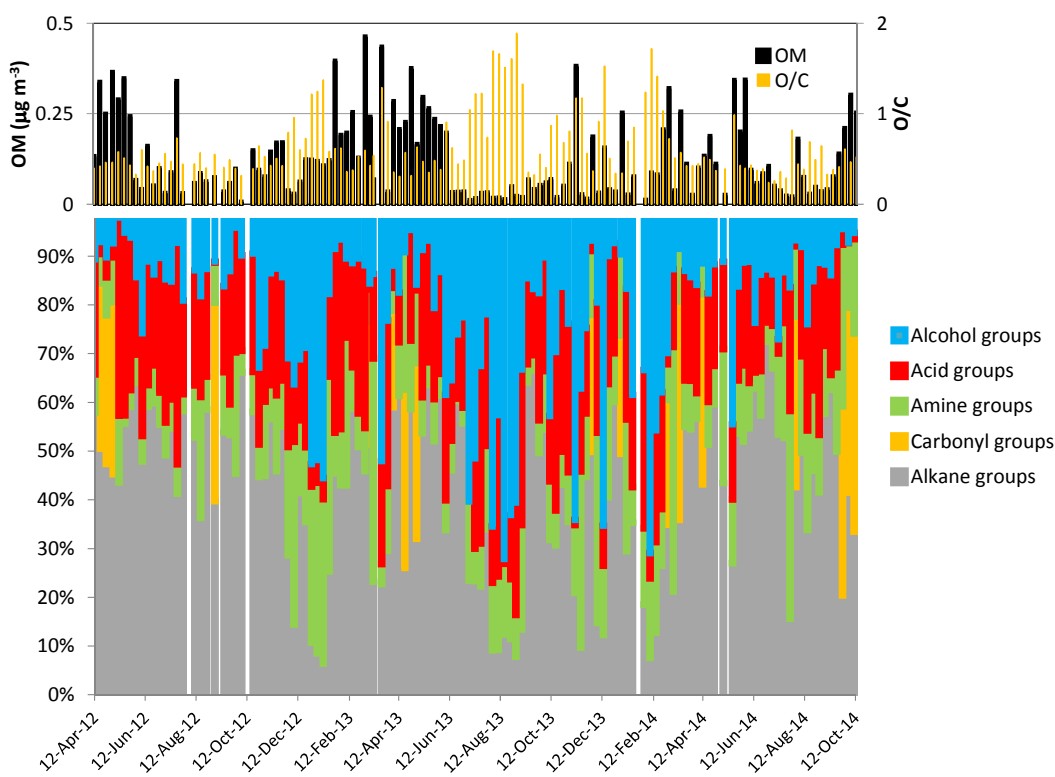

**Figure 4. Time series of weekly-averaged OM and O/C (top panel) and percent contributions of the functional groups to OM (bottom panel).**



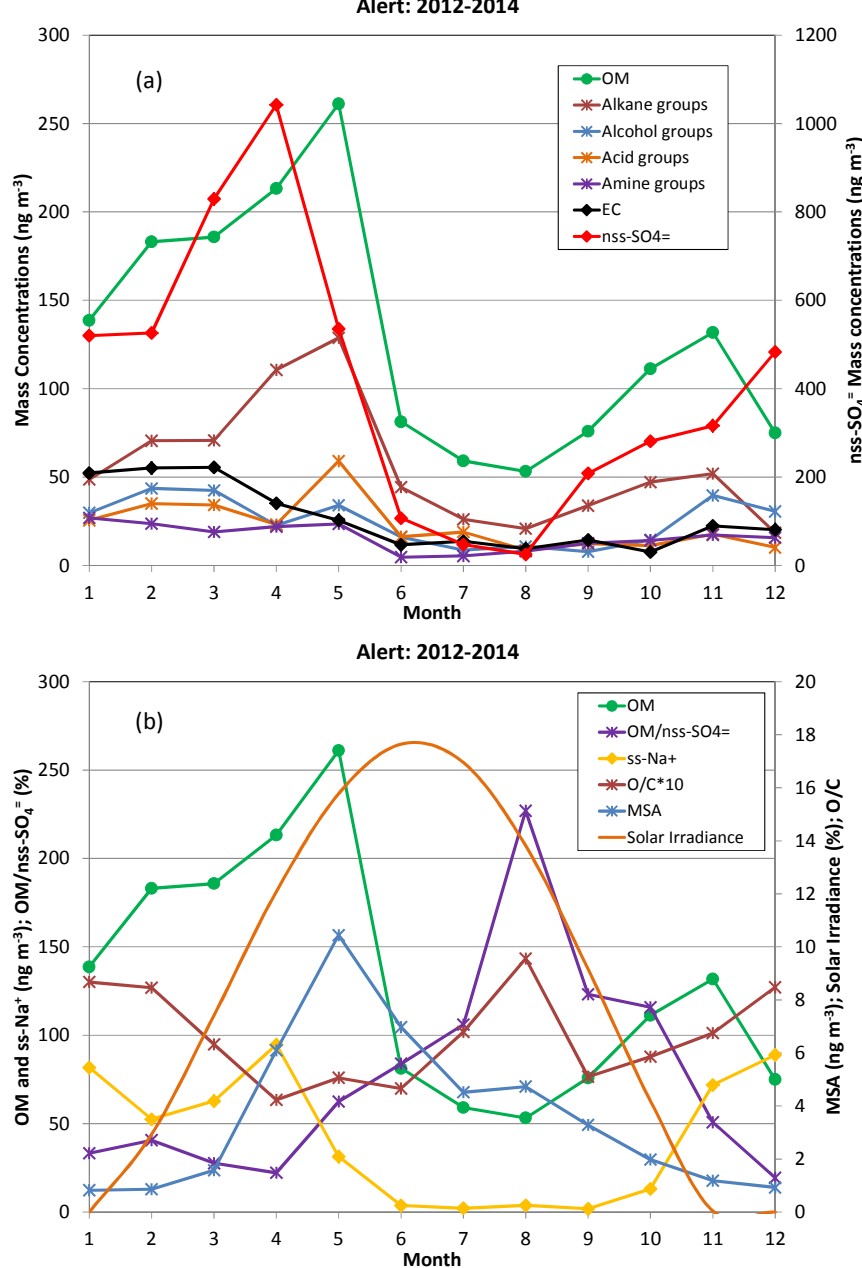

**Figure 5a and 5b. Monthly averages showing the annual variations of (a) OM, OM/nss-SO$_4^=$, ss-Na$^+$, O/C (multiplied by 10) and MSA, and (b) OM, functional group concentrations, EC and nss-SO$_4^=$. Also shown in (a) is the annual pattern of solar irradiance as a percentage of the total irradiance across a year.**





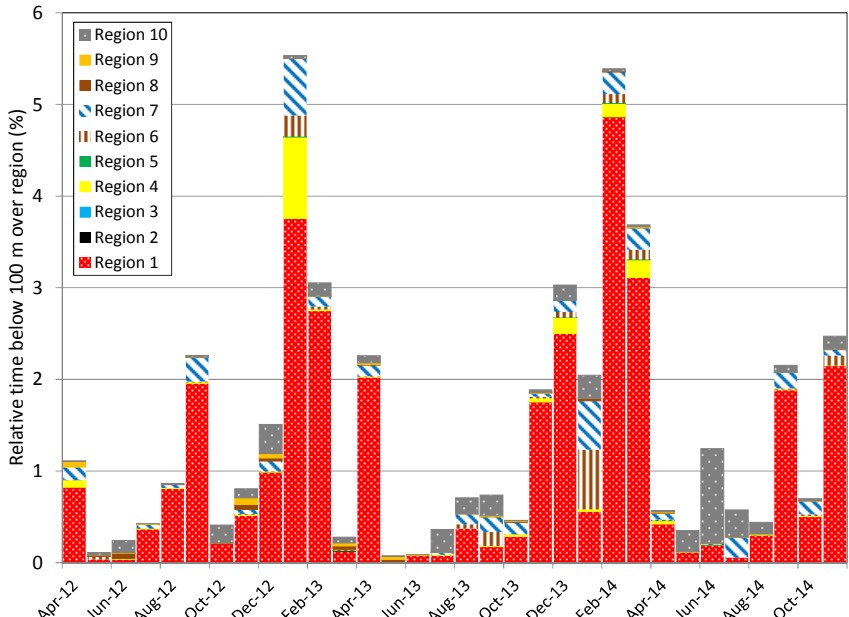

**Figure 6. Monthly-averaged percentage of previous 10 days spent below 100 m over Regions 1-10 identified in Fig. 1. Results are from the FLEXPART trajectory analyses for daily air parcels arriving at Alert.**





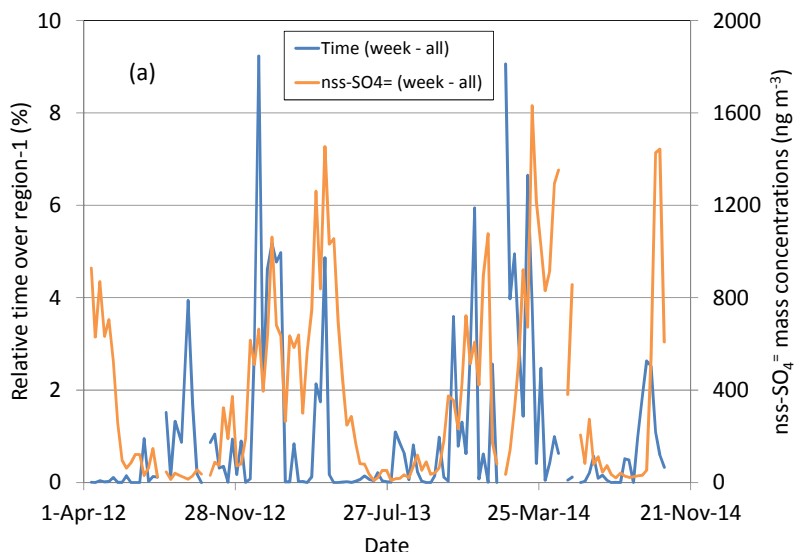

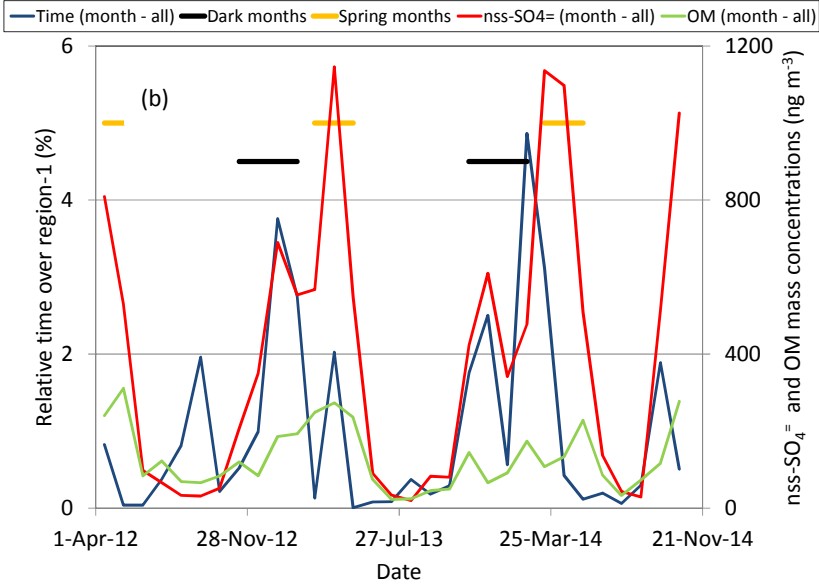

**Figure 7a and 7b. (a) Time series of percentage time over Region 1 from weekly averages of FLEXPART analysis with weekly-averaged nss-SO$_4^=$ mass concentrations. (b) Time series of time over Region 1 from monthly averages of FLEXPART analysis with monthly-averaged nss-SO$_4^=$ and OM concentrations. The bars in (b) delineate the dark months (NDJF) and spring months (MAM).**




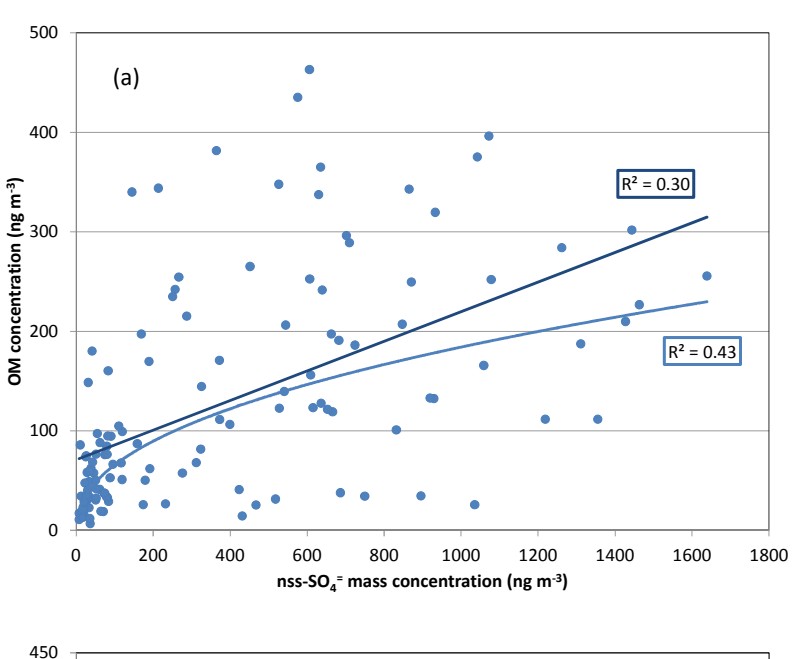

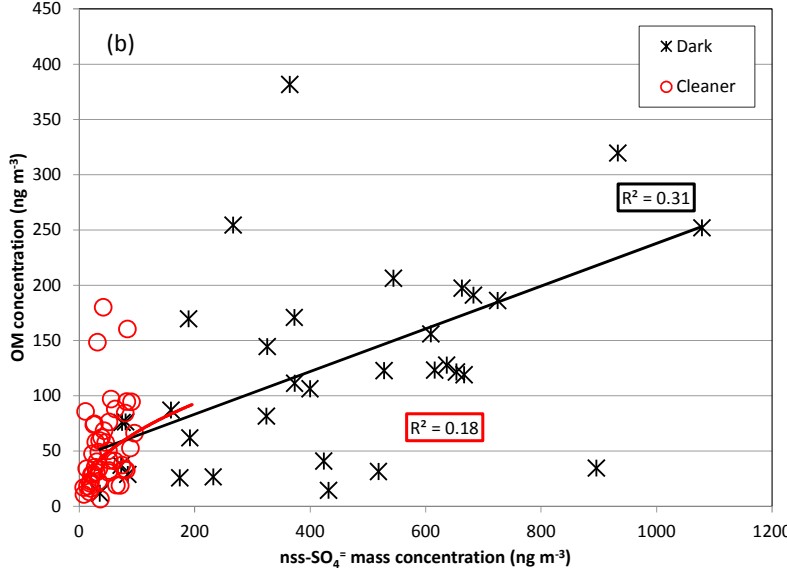

**Figure 8. Regressions of OM versus nss-SO$_4^=$ for a) all weeks and b) for dark weeks (during NDJF) and cleaner weeks (nss-SO$_4^=$ <100 ng m$^{-3}$). Coefficients of determination are indicated. Linear and power-law regressions are shown for all points (p<0.01), and linear regressions for the dark period (p<0.01) and the cleaner period (p<0.03).**




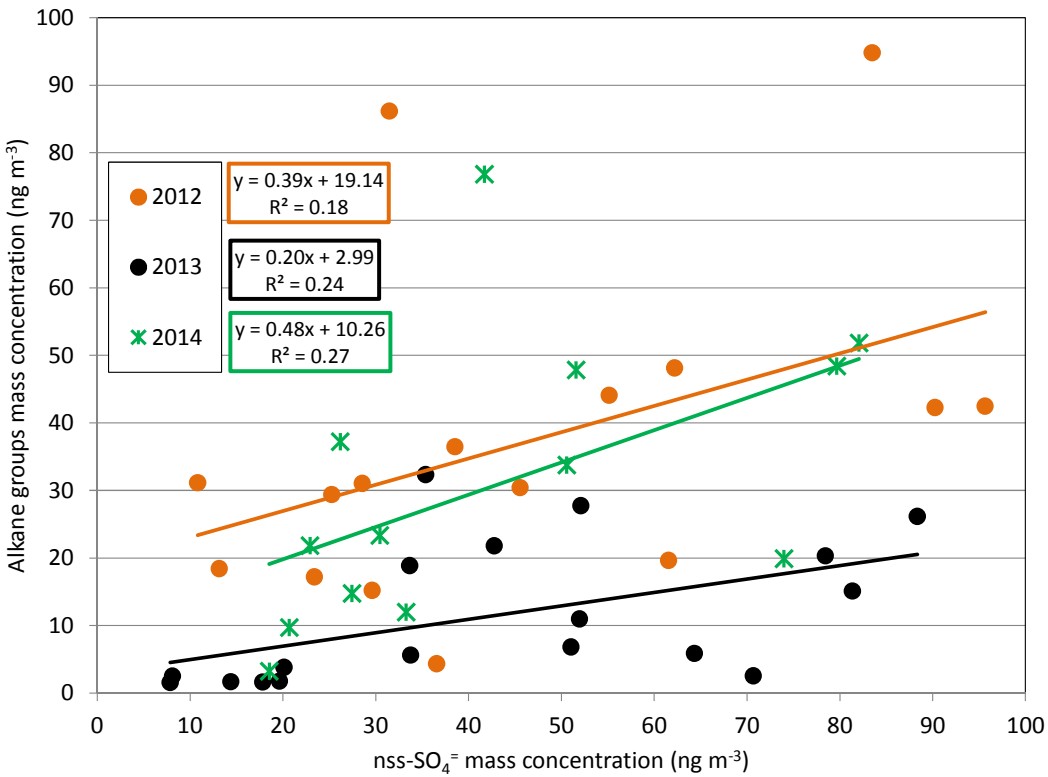

**Figure 9. Regressions of mass concentrations of alkane functional groups with nss-SO$_4^=$ for the cleaner weeks during each of 2012, 2013 and 2014.  (2012: p<0.1; 2013: p<0.04; 2014: p<0.07)**





**Figure 10.** Regressions of weekly-averaged mass concentrations of alcohol functional groups with ss-Na+ for the spring weeks, dark weeks, cleaner weeks and for the 10 dark weeks with O/C>1. Results of linear regressions are indicated where significant (P<0.05). Lower plot expands the cleaner weeks and identifies the nine weeks during July and August of 2013 with O/C>1.