# Peer review of "Organic Functional Groups in the Submicron Aerosol at 82.5°N, 62.5°W from 2012 to 2014"

_Atmospheric Chemistry and Physics, 2017_

## Referee Comment (RC1) · Anonymous Referee #3 · 2 Sep 2017

General comments This comprehensive study reports 2.5 years of weekly averaged data from the Arctic research observatory Alert. Measured species include elemental (EC) and organic carbon (OC), organic functional groups as measured by FTIR, inorganic species including oxalate and methanesulfonic acid (MSA), non-refractory species measured by an Aerosol Chemical Speciation Monitor (ACSM). Furthermore, particle size distributions were analysed by a Scanning Mobility Particle System (SMPS) and an Optical Particle Counter (OPC). The authors used chemical speciation in the Arctic aerosol, PMF and linear regression based on this data, and the transport model FLEXPART to associate organic aerosol components with source types and source regions. The manuscript is well written with sound discussions.

The large number of figures in the manuscript could be reduced, e.g. moved to the

supplementary section.

The authors state that filter-OC data will be published elsewhere, but it will add great value to the manuscript and be meaningful to compare this data with ACSM-OM con-centrations and OFG-OM that has a central position in the source attribution.

The authors build the factor analysis on organic functional groups, which may undergo atmospheric degradation during transport to Alert, which they correctly state could be much longer than 10 days. The authors should consider if chemical transformation of the functional groups into more oxidized oxidized in the (often acidic) arctic aerosols, could lead to erroneous conclusions in the PMF analysis. . Specific comments Line 181: How does the OFG-OM and ACSM-OM based mass concentrations compare with the offline OC measurements (estimating O/C equals that obtained from the FTIR measurements), and does the uncertainty in OFG-OM agree with that of Russel, 2003?

Line 205: How is the collection efficiency (CE) justified? Based on the variation in chemical composition over the year, in particular with respect to sulphate, the acidity could vary substantially. Thus, a variable CE would probably be appropriate, e.g. based on the parameterization method. More in Middlebrook et al., 2012. Line 520: Does the possible marine influence of Factor 1 agree with MSA, or is this stronger in Factor 3? Line 561: Does the OFG-OM agree with filter OC derived concentrations?

---

## Referee Comment (RC2) · Anonymous Referee #1 · 15 Oct 2017

Summary and Overall Recommendation:

Multi-year studies, such as the one presented here, on the chemical composition of submicron aerosol are highly needed in the literature and I feel are at times under-appreciated by the atmospheric chemistry community. This study is highly unique in that it presents multi-year data (i.e., April 2012 to October 2014) on the organic functional groups (OFGs) found in submicron aerosol collected from the Arctic at the Alert Observatory. A lot of important observations are made about OFGs in submircon aerosol collected from the Arctic during this study. For example, the authors found that a secondary marine source is likely a general feature of summer OM, but during years where there is likely more combustion-related sources (such as biomass burning) the contribution of alkane groups to the OM can be higher. Overall, I think this study will

be publishable in ACP. However, I do a few comments below that the authors should consider before full publication is considered.

1.) Generally, I feel at times the text in the discussion section can be a bit dense and hard to follow. I couldn't think of an easier way to reorganize the text, but I just thought to point this out to the authors, especially if this comment concerned them enough to consider reorganizing the discussion section.

2.) Abstract, Line 12: Change "lower organic mass concentrations (OM)" to "Lower organic mass (OM) concentrations"

3.) Abstract, Lines 16-17: If you are going to list the initial of the month in parentheses after each season, shouldn't you also do this for summer to be consistent?

4.) Abstract, Line 25: change "most persistence" to "most persistent"

5.) 2.1 Instrumental Methods, Page 4: What was the temperature of the freezer at the Observatory? This is important to know so that readers can judge if potential changes in composition might have occurred.

6.) 2.1 Instrumental Methods, Page 5: When the authors state "Prior to OFG analysis by FTIR spectroscopy, the filters were equilibrated in a temperature and humidity-controlled cleanroom environment for 24 h," what do you mean exactly? Is the temperature and RH always the same for all samples measured? What is the temperature and RH of this room? Does this change the composition since the aerosol were likely collected at much colder conditions in the Arctic?

7.) 2.1 Instrumental Methods, Page 5:

Did the authors consider conducting ammonium sulfate calibrations with the ACSM? Budisulistiorini et al. (2014, AMT) found this was necessary with the multi-year measurements of submicron aerosol in the southeastern U.S.

8.) Related to # 7 above, why didn't the authors consider presenting and comparing

OFG data with PMF analysis of ACSM OM? Was it that the signals were too low for PMF analyses?

9.) For OFG analyses by FTIR, one thing that really never comes across are the uncertainties of this technique, especially at low mass concentrations. Are the uncertainties accurately estimated? I worry that FTIR may have issues at these lower mass concentrations, which can affect all of the downstream analyses you conduct in this manuscript. This also relates to how well your peak-fitting method really works when you are limited by low amounts of OM collected on these filters. Are you missing any important functional groups? I would think offline mass spectral analyses of these filters should be something the authors consider in the future (not in this manuscript of course).

10.) When you say a "Mixed" factor this is very confusing to me. Is this mixed because PMF fails to resolve this potentially mixed statistical solution? I wonder if this is the case due to the robustness of OFG analysis by FTIR. As the authors now, this OFG analysis is not as specific as mass spec in resolving finer details in the chemistry. I guess this "mixed" factor is resulting from this underlying issue with OFG analysis by FTIR.

11.) I'm intrigued by the authors potential observation of secondary marine OM source. Do the authors think this could be BVOC-related emissions from plankton?

---

## Author Comment (AC1) · 2 Dec 2017

Responses We are grateful to both reviewers for their generous and constructive comments as well as their time. In the revised manuscript, all revisions are highlighted. Those based on specific comments are highlighted in yellow. Revisions to improve the discussion are highlighted in blue.

Referee #1 Summary and Overall Recommendation:

Multi-year studies, such as the one presented here, on the chemical composition of submicron aerosol are highly needed in the literature and I feel are at times underappreciated by the atmospheric chemistry community. This study is highly unique in that it presents multi-year data (i.e., April 2012 to October 2014) on the organic functional

groups (OFGs) found in submicron aerosol collected from the Arctic at the Alert Observatory. A lot of important observations are made about OFGs in submicron aerosol collected from the Arctic during this study. For example, the authors found that a secondary marine source is likely a general feature of summer OM, but during years where there is likely more combustion-related sources (such as biomass burning) the contribution of alkane groups to the OM can be higher. Overall, I think this study will be publishable in ACP. However, I do a few comments below that the authors should consider before full publication is considered.

1.) Generally, I feel at times the text in the discussion section can be a bit dense and hard to follow. I couldn't think of an easier way to reorganize the text, but I just thought to point this out to the authors, especially if this comment concerned them enough to consider reorganizing the discussion section.

Response – We appreciate that the writing is a little "dense", and we thank both reviewers for their tolerance. We have edited some areas of the discussion (blue highlighted text) to try and make it a little more palatable, but it remains dense.

2.) Abstract, Line 12: Change "lower organic mass concentrations (OM)" to "Lower organic mass (OM) concentrations"

Response – Done.

3.) Abstract, Lines 16-17: If you are going to list the initial of the month in parentheses after each season, shouldn't you also do this for summer to be consistent?

Response – It is not given here because it was done on line 13.

4.) Abstract, Line 25: change "most persistence" to "most persistent"

Response – Corrected.

5.) 2.1 Instrumental Methods, Page 4: What was the temperature of the freezer at the Observatory? This is important to know so that readers can judge if potential changes

in composition might have occurred.

Response – The freezer temperature (measured at -18.5C) is now given on line 35 of page 4.

6.) 2.1 Instrumental Methods, Page 5: When the authors state "Prior to OFG analysis by FTIR spectroscopy, the filters were equilibrated in a temperature and humidity controlled cleanroom environment for 24 h," what do you mean exactly? Is the temperature and RH always the same for all samples measured? What is the temperature and RH of this room? Does this change the composition since the aerosol were likely collected at much colder conditions in the Arctic?

Response - The filters are stored in sealed petri dishes inside double zip loc bags while frozen (below 0C). The closed petri dishes are moved from the freezer into the cleanroom for 24 hr prior to measurement in the FTIR spectrometer. The cleanroom is maintained at 20C and <40% relative humidity. A continuous N2 purge is used in the spectrometer. There is no evidence that the composition changes measurably during the freezer storage or cleanroom equilibration. This discussion has been added to lines 4-7 of page 5.

7.) 2.1 Instrumental Methods, Page 5: Did the authors consider conducting ammonium sulfate calibrations with the ACSM? Budisulistiorini et al. (2014, AMT) found this was necessary with the multi-year measurements of submicron aerosol in the southeastern U.S.

Response – Since we first started using the ACSM in 2009 (Takahama et al., ACP, 2011; Leaitch et al., AE, 2011), we have done spans with ammonium sulphate. We identified to Aerodyne a large difference in the measurement of sulphate by the ACSM compared with the old quad AMS and a HR AMS in 2010. The sulphate data here use the new relative ionization efficiency for sulphate. In association we have added reference to Budisulistiorini et al. (2014, AMT) on page 5, lines 35-36. We have had larger problems that have been hampered by limited hands-on attention at Alert and quite

slow remote connections. With our limited personnel, ACSM software and hardware problems as well as upgrades, we have struggled to keep the instruments providing serviceable data; hence the limitations on the ACSM data here.

8.) Related to # 7 above, why didn't the authors consider presenting and comparing OFG data with PMF analysis of ACSM OM? Was it that the signals were too low for PMF analyses? Response – For the reasons mentioned in response to #7, our ACSM data are limited. We did considered using the few months of ACSM data in a more direct sense. However, compared with filter datasets covering approximately two and a half years, we believe it would substantially increase the complexity of the discussion without a clear advantage, and particularly considering the uncertainty in the CE.

9.) For OFG analyses by FTIR, one thing that really never comes across are the uncertainties of this technique, especially at low mass concentrations. Are the uncertainties accurately estimated? I worry that FTIR may have issues at these lower mass concentrations, which can affect all of the downstream analyses you conduct in this manuscript. This also relates to how well your peak-fitting method really works when you are limited by low amounts of OM collected on these filters. Are you missing any important functional groups? I would think offline mass spectral analyses of these filters should be something the authors consider in the future (not in this manuscript of course).

Response - Since samples were collected for multiple days in order to obtain mass loadings of 10 to 50 ug, comparable to studies in polluted areas with 4-6 hr sample times, the detection limits reported previously [Gilardoni et al.; Maria et al.] were applied here and the resulting uncertainties are similar to those previously reported (+/-21% for OM [Russell, 2003]). Not all OFG were above detection, as reported in the manuscript. Their absence is included in the estimated uncertainty, as discussed by Russell [2003].

10.) When you say a "Mixed" factor this is very confusing to me. Is this mixed because

PMF fails to resolve this potentially mixed statistical solution? I wonder if this is the case due to the robustness of OFG analysis by FTIR. As the authors now, this OFG analysis is not as specific as mass spec in resolving finer details in the chemistry. I guess this "mixed" factor is resulting from this underlying issue with OFG analysis by FTIR.

Response - In studies where more samples are taken at higher time resolution, more PMF factors have been resolved by FTIR [e.g. S. Liu et al. 2012], so the limitation is not the analysis technique but the number of samples. The number of samples is limited at Alert both by the clean conditions and the limited access and resources.

11.) I'm intrigued by the authors potential observation of secondary marine OM source. Do the authors think this could be BVOC-related emissions from plankton?

Response – We previously referenced papers by Willis (page 12, line 33), Mungall (page 14, line 16), Facchini (page 13, line 17) and Köllner (page 13, line 17; page 15, line 9) concerning secondary marine sources. We prefer to leave it at those references, rather than speculate here on the nature of secondary marine OM in the Arctic. Mungall et al found no evidence of significant levels of isoprene and monoterpenes in the gasphase over the waters of Baffin Bay and the Nares Strait.

---

## Author Comment (AC2) · 2 Dec 2017

Responses We are grateful to both reviewers for their generous and constructive comments as well as their time. In the revised manuscript, all revisions are highlighted. Those based on specific comments are highlighted in yellow. Revisions to improve the discussion are highlighted in blue.

Referee #3

General comments This comprehensive study reports 2.5 years of weekly averaged data from the Arctic research observatory Alert. Measured species include elemental (EC) and organic carbon (OC), organic functional groups as measured by FTIR, inorganic species including oxalate and methanesulfonic acid (MSA), non-refractory

species measured by an Aerosol Chemical Speciation Monitor (ACSM). Further-more, particle size distributions were analysed by a Scanning Mobility Particle System (SMPS) and an Optical Particle Counter (OPC). The authors used chemical speciation in the Arctic aerosol, PMF and linear regression based on this data, and the transport model FLEXPART to associate organic aerosol components with source types and source regions. The manuscript is well written with sound discussions. The large number of figures in the manuscript could be reduced, e.g. moved to the supplementary section.

The authors state that filter-OC data will be published elsewhere, but it will add great value to the manuscript and be meaningful to compare this data with ACSM-OM concentrations and OFG-OM that has a central position in the source attribution. The authors build the factor analysis on organic functional groups, which may undergo atmospheric degradation during transport to Alert, which they correctly state could be much longer than 10 days. The authors should consider if chemical transformation of the functional groups into more oxidized in the (often acidic) arctic aerosols, could lead to erroneous conclusions in the PMF analysis.

Response 1) We do not wish to make this an intercomparison paper, but we have revised Supplement Figure S2 to include a time series comparison of OC from the Thermal Method (TM; details added to the Methods of the main text) and OC from the OFG analyses as S2a. In Figure S2c, we show a regression of the OFG-OC and TM-OC. A description of the TM-OC has been added to the Methods (page 4, lines 23-27). The TM-OC is based on OC derived at 550oC plus OC derived at 870oC. The additional OC at 870oC is based on isotopic analyses showing no significant carbonate (no reference). OC at 870oC has been shown before to correlate with OC and water soluble OC (Chan et al., ACP, 2010). For corresponding points, the mean TM-OC, OFG-OC and OFG-OM are 120 ng/m3, 64 ng/m3 and 120 ng/m3, respectively (now discussed on page 6, lines 10-14). Where the TM-OC is higher than the OFG-OM in the summer of 2013, the O/C calculated from the OFG co-varies with mz44/mz43 from

the ACSM. Also, the OFG-OM, shown in Figure S2b, compares reasonably with OM estimated from the ACSM (see response 4 below). On factor that may contribute to the higher OC by the TM method is absorption of VOCs by the quartz filters used for sampling. Overall, we believe the variations in O/C, estimated from the OFG analysis, to be reasonable within defined uncertainities.

Response 2) We have added a statement (page 14, lines 7-8) to the effect that these OFG may have undergone transformation during transport, and that may impact the PMF analysis. However, whether and how this might impact the PMF analysis is unclear.

Specific comments

Line 181: How does the OFG-OM and ACSM-OM based mass concentrations compare with the offline OC measurements (estimating O/C equals that obtained from the FTIR measurements), and does the uncertainty in OFG-OM agree with that of Russel, 2003?

Response 3) – Please see Response 1 above. It seems that with this comment the reviewer is implying we can assess the uncertainty in the OFG-OM based on the ACSM-OM and the uncertainty in OFG-OC based on the TM-OC measurements. Whether that is the implication or not, the uncertainty in the ACSM-OM is relatively large due to the CE (see Response 4 below), and the TM-OC uncertainty is approximately +/- 25%. Based on +/-25% for each of OFG-OC and TM-OC, their means fall within the uncertainty range. That has been added on page 7, lines 9-13.

Line 205: How is the collection efficiency (CE) justified? Based on the variation in chemical composition over the year, in particular with respect to sulphate, the acidity could vary substantially. Thus, a variable CE would probably be appropriate, e.g. based on the parameterization method. More in Middlebrook et al., 2012.

Response 4) – We calculated the CE based on equations 4 and 7 of Middlebrook et al. (2012). The resulting values of CE were relatively high (e.g. 0.75), which resulted in

concentrations of sulphate and organics very much lower than the filter concentrations (either OFG or TM). Assuming that the organics are internally mixed with sulphate in the particles measured at Alert, we took the approach of Quinn et al. (2006) and set the CE based on a comparison of the raw ACSM SO4 with the filter sulphate. The average of the CE values calculated that way is 0.20, but, as we show, the filter sulphate may include sulphate in particles larger than efficiently sampled by the ACSM (>500 nm). We also calculated ACSM-OM using a constant CE of 0.5 that is commonly used in many AMS publications and is consistent with the organic aerosol discussion of Midddlebrook et al (2012). Time series of OFG-OM, TM-OC, ACSM-OM (CE based on comparison with filter sulphate and for a constant CE of 0.5) for the period of ACSM data are shown in Figure S2b. The statistics of linear regressions of OFG-OM with ACSM-OM and TM-OC with ACSM-OM, for the sulphate-based CE, are given in the caption, and they show that the OFG-OM and the ACSM-OM compare within viable ACSM collection efficiencies. This discussion has been added on page 6, lines 24-33.

Line 520: Does the possible marine influence of Factor 1 agree with MSA, or is this stronger in Factor 3?

Response 5) – The correlation with MSA is highest for factor 3. For factor 1, the CoD is 0.02. We have added the following sentence to the end of the paragraph discussing factor 1 (page 14, lines 19-20): "However, there are no correlations of this factor with Na+ or MSA."

Line 561: Does the OFG-OM agree with filter OC derived concentrations?

Response – See response to 1.

---

## Author Response (AR2)

To the Co-Editor – Thank you for your help and time with this manuscript. With the exception of one highlighted comment below, all corrections have been made. The corrections are highlighted in yellow in the manuscript.

Co-Editor Decision: Publish subject to minor revisions (review by editor) (21 Dec 2017) by Willy Maenhaut

Comments to the Author:

The authors have reasonably addressed the comments of the two anonymous referees and they have modified their manuscript accordingly. However, the comments given below should be addressed and several alterations are needed for the Main text and Supplement before the manuscript can be published in ACP.

Main text:

Page 1, line 1: Since the study pertains to the Alert station, I suggest to insert ", 62.5°W" after "82.5°N" in the title.

Page 1, line 15, and also further in the manuscript (e.g., in Table 1): Concentration data starting with "2" or higher as first significant figure should be provided with at most 2 significant figures; in case the first significant figure is "1", 3 significant figures can be used.

Page 2, lines 4 and 5, and throughout the remainder of the text: Replace "e.g. " by "e.g., ".

Page 2, line 13: Replace "Stohl (2006)" by "Stohl, 2006".

Page 3, line 20: Replace "Zeppelin, indicates" by "Zeppelin indicates".

Page 3, line 26: Replace "5% of" by "5% of the".

Page 3, line 33: Replace "75°W" by "62.5°W".

Page 4, line 19: Replace "Teflon for" by "Teflon filters for".

Page 4, lines 21-22: Delete ", also sampled at 27 L min-1," as the flow rate was already mentioned in line 19.

Page 5, line 4: Replace "0C" by "0°C".

Page 5, line 6: Replace "20C" by "20°C".

Page 5, line 6: The "2" of "N2" should be in subscript.

Page 5, line 29: The "μ" character after "to 10 " did not come out properly in the pdf file.

Page 5, line 34: Replace "One hour" by "One-hour".

Page 5, line 36: Replace "AMT, 2014" by "2014".

Page 7, line 3: It is not clear to what the OM in equation (1) refers. This should be clarified. I suspect that it refers to the OFG-OM.

Page 8, line 20: Replace "i.e. winter" by "i.e., winter".

Page 10, line 1: Replace "differences" by "difference". Response – Because the difference being referred to is among 3 summers, I believe it is correct as "differences".

Page 10, line 3: Replace "3.3 – Potential" by "3.3 Potential".

Page 10, line 12: Abbreviations or acronyms, here "BB", should be defined (written full-out) when first used within the main text after the Abstract.

Page 10, lines 16-19: The wording in the second part of this sentence starting with "and times over the dominant" is strange and unclear to me.

Page 10, line 19: It is unclear which measures are meant by "the same two measures".

Page 12, line 13: Abbreviations or acronyms, here "DL", should be defined (written full-out) when first used within the text.

Page 14, line 4: Replace "3.5 – Positive" by "3.5 Positive".

Page 15, line 23: Replace "occurs during" by "occur during".

Page 16, line 19: Replace "BB" by "biomass burning (BB)".

Page 16, line 21: Replace "respectively were" by "respectively, were".

Page 16, line 29: Replace "biomass burning (BB)" by "BB".

Page 18, caption of Fig. 5: The caption disagrees with what is shown. Figures (a) and (b) should be exchanged.

Page 18, line 33: "for all" by "for a) all".

Pages 19-28, References: Titles of journal articles should be in lower case instead of in Title Case.

Page 21, lines 17-30: "Fu, P., et al., 2015" should come before "Fu, P. Q., et al., 2009a".

Page 21, line 24: Replace "Fu, P. K." by "Fu, P. Q.".

Page 24, line 5: Replace "R. and Deschler" by "R., and Deschler".

Page 25, lines 33-37: "Quinn et al., 2007" should come before "Quinn et al., 2009".

Page 29, heading of Table 1: Replace "Mass Concentrations" by "mass concentrations".

Page 29, footnote of Table 1: Replace "(BDL)" by "(DL)" and replace "carbonyl BLD" by "carbonyl below DL".

Page 29, heading of Table 2: "group-2 regions" is confusing and can in my opinion simply be replaced by "regions". Furthermore, replace "in brackets" by "in parentheses".

Page 30, heading of Table 3: Replace "of Determination for Linear Regressions" by "of determination for linear regressions".

Supplement:

Caption of Figure S1: Replace "station Influence" by "station influence".

Caption of Figure S4: Replace "region 11 (Canadian NWT)" by "Canadian NWT region" as there is no region 11 in Figure 1.

Heading of Table S1: Replace "in brackets" by "in parentheses".

---

## Author Response (AR3)

Thank you for pointing this out.  The change on page 7 has been made.